# Blunted diurnal firing in lateral habenula projections to dorsal raphe nucleus and delayed photoentrainment in stress-susceptible mice

He Liu[1,2,3☯], Ashutosh Rastogi[1☯], Priyam Narain[4], Qing Xu[4], Merima Sabanovic[1], Ayesha Darwish Alhammadi[1], Lihua Guo[4], Jun-Li Cao[2,4], Hongxing Zhang[2], Hala Aqel[1], Vongai Mlambo[1], Rachid Rezgui[1], Basma Radwan[1], Dipesh Chaudhury[1]*

1 The Division of Science, New York University Abu Dhabi, Abu Dhabi, United Arab Emirates, 2 Department of Anesthesiology, The Affiliated Hospital of Xuzhou Medical University, Xuzhou, China, 3 Jiangsu Province Key Laboratory of Anesthesiology & Jiangsu Province Key Laboratory of Anesthesia and Analgesia Application Technology, The Xuzhou Medical University, Xuzhou, China, 4 Center for Genomics and Systems Biology, New York University Abu Dhabi, United Arab Emirates

☯ These authors contributed equally to this work.
* dc151@nyu.edu

**Data Availability Statement:** All relevant data are within the paper and its files.

## Abstract

Daily rhythms are disrupted in patients with mood disorders. The lateral habenula (LHb) and dorsal raphe nucleus (DRN) contribute to circadian timekeeping and regulate mood. Thus, pathophysiology in these nuclei may be responsible for aberrations in daily rhythms during mood disorders. Using the 15-day chronic social defeat stress (CSDS) paradigm and in vitro slice electrophysiology, we measured the effects of stress on diurnal rhythms in firing of LHb cells projecting to the DRN (cells$^{LHb \rightarrow DRN}$) and unlabeled DRN cells. We also performed optogenetic experiments to investigate if increased firing in cells$^{LHb \rightarrow DRN}$ during exposure to a weak 7-day social defeat stress (SDS) paradigm induces stress-susceptibility. Last, we investigated whether exposure to CSDS affected the ability of mice to photoentrain to a new light–dark (LD) cycle. The cells$^{LHb \rightarrow DRN}$ and unlabeled DRN cells of stress-susceptible mice express greater blunted diurnal firing compared to stress-näive (control) and stress-resilient mice. Daytime optogenetic activation of cells$^{LHb \rightarrow DRN}$ during SDS induces stress-susceptibility which shows the direct correlation between increased activity in this circuit and putative mood disorders. Finally, we found that stress-susceptible mice are slower, while stress-resilient mice are faster, at photoentraining to a new LD cycle. Our findings suggest that exposure to strong stressors induces blunted daily rhythms in firing in cells$^{LHb \rightarrow DRN}$, DRN cells and decreases the initial rate of photoentrainment in susceptible-mice. In contrast, resilient-mice may undergo homeostatic adaptations that maintain daily rhythms in firing in cells$^{LHb \rightarrow DRN}$ and also show rapid photoentrainment to a new LD cycle.

**Funding:** The authors have received funding from the following sources: NYUAD Start-Up Fund (DC), NYUAD Annual Research Fund (DC), NYUAD Research Enhancement Fund (DC), University Research Challange Fund (DC), NARSAD (22715; DC), Al Jalila Research Foundation (AJF201638; DC), National Natural Science Foundation of China (NSFC81300957: HL), The Natural Science Foundation of Jiangsu Province (BK20181145: HL), Research Start-up Funding for Talent Introduction (2019203002: HL) and Clinical Technical Research and Study Plan Project (2018211006: HL). The funders had no role in study design, data collection and analysis, decision to publish, or preparation of the manuscript.

**Competing interests:** The authors have declared that no competing interests exist.

**Abbreviations:** aCSF, artificial cerebrospinal fluid; CRF, corticotrophin releasing factor; CSDS, chronic social defeat stress; DD, complete darkness; DR, delayed rectifier Kv+; DRN, dorsal raphe nucleus; EPSP, excitatory postsynaptic potential; HCN, hyperpolarization-activated cation; IKA, inactivation A-type Kv+; LD, light–dark; LHb, lateral habenula; p-GSK-3β, phosphorylated glycogen synthase kinase beta; PK2, prokineticin 2; SCN, suprachiasmatic nucleus; SDS, social defeat stress; SI, social interaction; SIK1, salt-inducible kinase 1; SSDS, subthreshold social defeat stress; TEA, tetraethylammonium; ZT, Zeitgeber time.

## Introduction

Mood disorders are associated with abnormalities in circadian rhythms in physiology and behaviour. The lateral habenula (LHb) and dorsal raphe nucleus (DRN) regulate diverse behaviours associated with mood, including cognition, reward, and sleep–wake cycle [1,2]. Moreover, observations that the LHb and DRN are connected to the suprachiasmatic nucleus (SCN), the master circadian clock, and that both nuclei exhibit diurnal rhythms in clock genes expression and neuronal activity [1,3] highlights the possible evolutionarily significant relationship between circuits that regulate motivational behaviours and the daily rhythms in such behaviours [2]. It is hypothesized that pathophysiological changes in these circuits may be responsible for the pathogenesis of psychiatric disorders and associated changes in behavioural rhythms [2,4].

Given the overlapping influence of the LHb and DRN on circadian rhythmicity and mood disorders, we assessed whether exposure to chronic social defeat stress (CSDS) affects: (i) diurnal rhythmic activity of LHb cells projecting to the DRN (cells$^{LHb \to DRN}$); (ii) DRN cells; and (iii) the ability of mice to reentrain to a new photoperiod. We demonstrate that cells$^{LHb \to DRN}$ and DRN cells of stress-susceptible mice exhibit blunted daily rhythms in neural activity. Specifically, the firing rate of cells$^{LHb \to DRN}$ was elevated in both, the day and night, in susceptible-mice, while resilient and stress-naïve mice display robust diurnal rhythms in firing with high activity in night and low in daytime. Moreover, the blunted diurnal rhythms in firing in cells$^{LHb \to DRN}$ of susceptible mice was inverted relative to control and resilient mice. In contrast, the firing rate of DRN cells was equally low in the day, while control and resilient mice exhibit high firing in the day compared to night. The blunted rhythms in firing rate in stress-susceptible mice is similar to observations that patients with mood disorders exhibit blunted daily rhythms in various physiological measures [5,6]. In addition to pathophysiological changes in firing in the cells$^{LHb \to DRN}$ and DRN cells, susceptible mice also exhibit delayed rate of reentrainment to a new light cycle, while resilient mice were faster at adapting to the new light cycle. Our findings suggest that chronic social stress induces pathophysiological firing in cells$^{LHb \to DRN}$ and DRN cells of susceptible mice that leads to blunted diurnal activity in this circuit. Moreover, the behavioural locomotor data suggest that molecular and cellular processes responsible for acute photoentrainment seem slower at responding to a new light stimulus in susceptible mice, while homeostatic adaptations in resilient mice increase their efficiency at adapting to the new light cycle.

## Results

### Increased daytime firing in LHb cells of susceptible mice

Though chronic stress leads to increased activity in the LHb [7], it is unclear whether the LHb of mice resilient or susceptible to strong stressors such as 15 days of CSDS exhibit similar increases in firing. To explore the firing properties in these mice, we performed in vivo, extra-cellular single-unit recordings in the daytime in anesthetized mice that had undergone CSDS (**Fig 1A–1C**; **S1A Fig**). Susceptible mice spent less time interacting with the social target as evidenced by the significantly lower social interaction ratio (**Fig 1D**). The spontaneous firing rate was significantly higher in susceptible mice relative to control (stress-naïve) and stress-resilient-mice (**Fig 1E**), which suggests that increased daytime activity in the LHb cells encode for susceptibility to CSDS. Though statistically not significant, there is a slight trend showing a negative correlation between firing rate and social interaction ratio (**S1B Fig**).

### Blunted rhythms in spontaneous firing in cells$^{LHb \to DRN}$ of susceptible mice

A majority of LHb cells express peak firing and increased c-Fos expression during the night [8,9]. Moreover, the LHb has reciprocal connections with the DRN [7], a region that

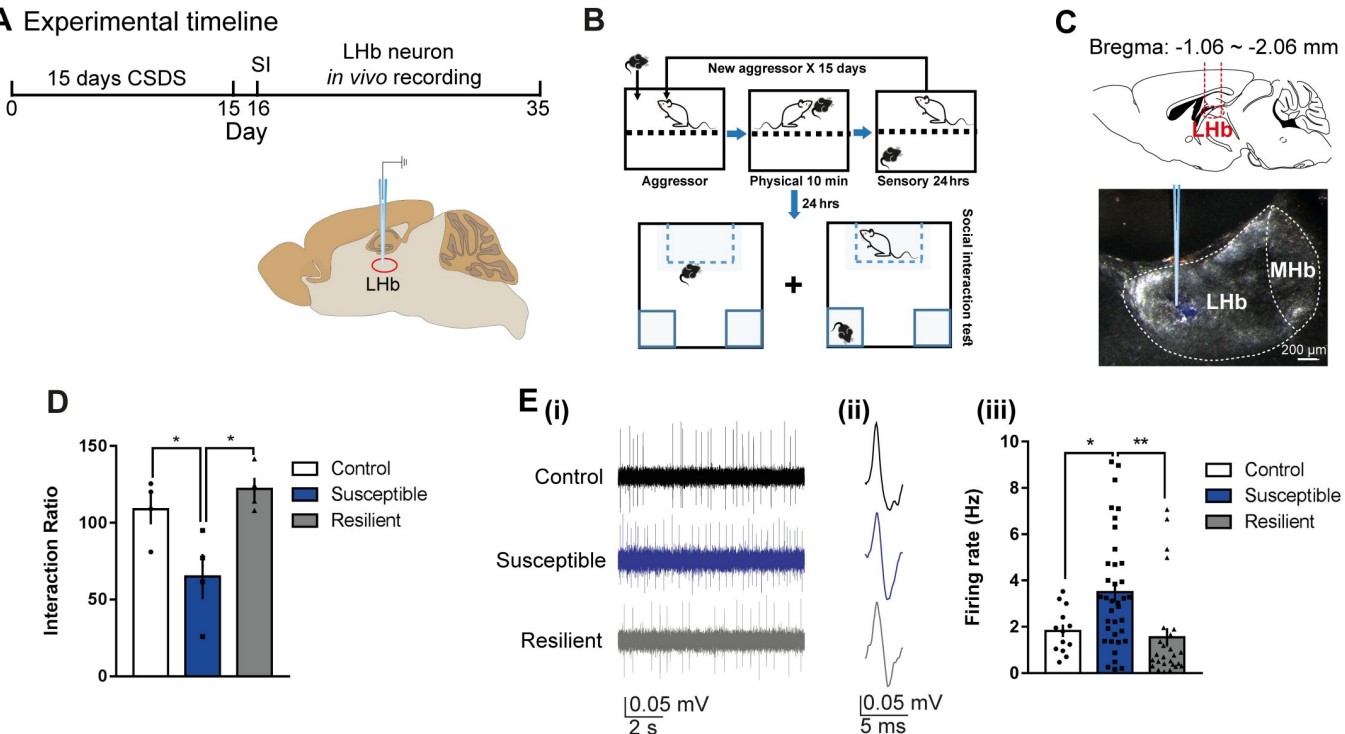

**Fig 1. Increased daytime firing in LHb cells of susceptible mice.** (A) Experimental timeline of 15 days CSDS paradigm, SI, and in vivo single-unit recording in the daytime. **(B)** Detailed schematic of the CSDS paradigm using C57BL/6J male mice and CD1 retired male breeders (aggressor) and SI test. **(C)** Schematic showing anatomical site of LHb (top) and example site of glass electrode recording in the LHb (bottom). **(D)** SI data showing that in the presence of a CD1 social target (nonaggressor), susceptible mice display decreased SI ratio ($F_{2,9}$ = 7.295, $P$ < 0.05, $N$ = 4 mice/group). **(E) (i)** Example traces, **(ii)** typical action potentials of recorded LHb cells of control (top), susceptible (middle), and resilient mice (bottom) after SI test. **(iii)** Increased firing rate in susceptible mice ($F_{2,73}$ = 7.195, $P$ = 0.0014; $N$ = 4 mice/group, $n$ = 13–37 cells/group). Error bars: mean ± SEM. The raw data can be found in S1 Data. CSDS, chronic social defeat stress; LHb, lateral habenula; MHb, medial habenula; SI, social interaction.

modulates sleep–wake states [10] and mood [11]. Thus, we questioned whether exposure to CSDS affected diurnal rhythmic activity in cells$^{LHb \rightarrow DRN}$. To explore this rhythmic characteristic of cells$^{LHb \rightarrow DRN}$, a combination of retrograding AAV5-Cre and conditional AAV-DIO-mCherry was used to specifically label cells$^{LHb \rightarrow DRN}$ followed by CSDS (**Fig 2A and 2B**). Immunohistochemical labelling showed that approximately 5% ± 0.1% of cells from the medial and lateral portion of the LHb (LHbM and LHbL) project to the DRN (**Fig 2C**). To explore the firing characteristic of these cells, the mice underwent CSDS followed by the social interaction (SI) test in order to determine the resilient and susceptible phenotypes. The mice were then euthanized in the day (ZT1) or the night (ZT13) followed by in vitro slice electrophysiological recordings of the fluorescently labelled cells$^{LHb \rightarrow DRN}$. The SI scores for control, resilient, and susceptible mice were comparable for animals where electrophysiological recordings were performed in the day or the night (**S2A and S2B Fig**). Daytime spontaneous firing rate of cells$^{LHb \rightarrow DRN}$ of susceptible mice was significantly higher compared to control and resilient mice (**Fig 2D**). In contrast, there was no significant difference in nighttime spontaneous firing rate in cells$^{LHb \rightarrow DRN}$ between control, resilient, and susceptible mice (**Fig 2E**). Moreover, the data show a significant negative correlation between firing rate and SI ratio during the day but not at night (**Fig 2F and 2G**). One-way ANOVA was performed to compare the spontaneous firing rates between the phenotypes in the day and the night as given in Fig 2D and 2E. Subsequent statistical analysis established a significant difference in firing between the day and night; F(5,97) = 5.5; $P$ = 0.0002. Post hoc analysis further confirmed that spontaneous daytime

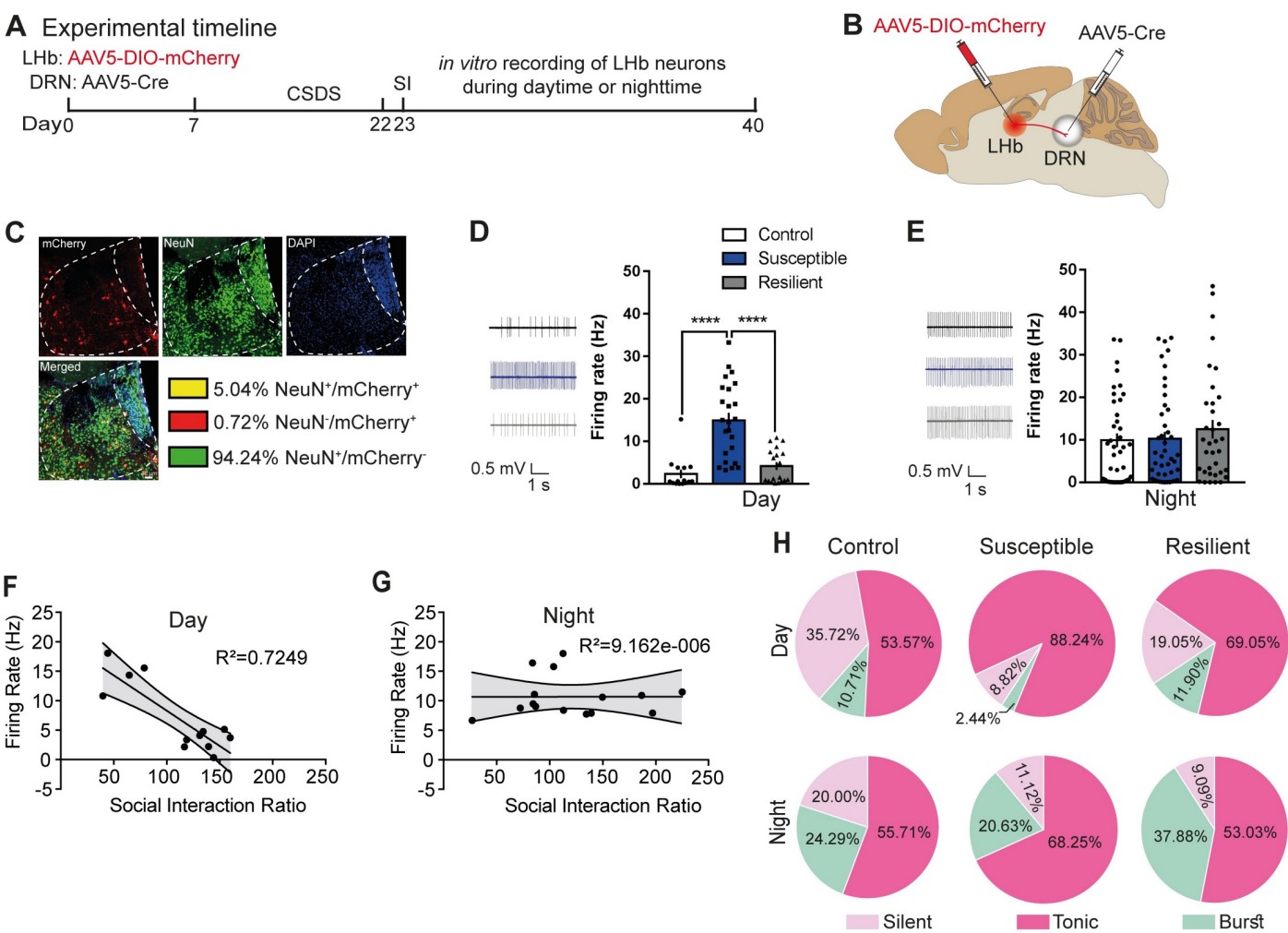

**Fig 2. Blunted rhythms in spontaneous firing in cells**$^{LHb\rightarrow DRN}$ **of susceptible mice.** (A) Experimental timeline of viral surgeries to label cells$^{LHb-DRN}$, CSDS, SI test, and in vitro slice electrophysiology. (B) Schematic showing surgeries for viral injections to specifically label cells$^{LHb-DRN}$. (C) Confocal images showing coexpression of mCherry, NeuN, and DAPI within LHb; and quantified data showing mCherry-expressing neurons (5% ± 0.1% LHb neurons project to DRN). (D) Sample traces of daytime in vitro spontaneous activity in cells$^{LHb-DRN}$ of mice. Increased daytime spontaneous activity in cells$^{LHb-DRN}$ in susceptible mice (right) ($F_{2,55} = 22.68$, $P < 0.0001$; $n = 15$–24 cells from 7 to 11 mice/group). (E) Sample traces of nighttime in vitro spontaneous activity in cells$^{LHb-DRN}$ of mice. No difference in nighttime spontaneous firing in cells$^{LHb-DRN}$ in control, susceptible, and resilient mice (right). (F, G) Significant negative correlation between spontaneous activity in cells$^{LHb-DRN}$ and SI ratio during the day ($R^2 = 0.7249$; $P < 0.05$) but not the night. (H) Pie charts illustrating the percentage of cells expressing: silent, tonic, or burst firing in the daytime (top) and nighttime (bottom) in labelled cells$^{LHb-DRN}$. Susceptible mice exhibit notable increased daytime tonic firing and decrease in silent patterns ($\chi^2 = 6.947$, $P < 0.05$, chi-squared test; $n = 28$–42 cells from 7 to 11 mice/group). At night, there is no difference in firing patterns among the phenotypes. There is significant increase in burst firing at night compared to day ($P < 0.05$). Error bars: mean ± SEM. The raw data can be found in S2 Data. CSDS, chronic social defeat stress; DRN, dorsal raphe nucleus; LHb, lateral habenula; SI, social interaction.

firing for control and resilient mice was significantly lower than spontaneous nighttime firing in control and resilient mice (control-day versus control-night: $P = 0.02$; resilient-day versus resilient-night: $P = 0.02$). In contrast, post hoc comparison between susceptible mice showed that spontaneous daytime firing was significantly higher than the night ($P = 0.008$). Replotting the data of the spontaneous firing in the first and second week of recordings showed that control and resilient mice exhibited robust differences in the pattern of diurnal activity where firing was lower in the day and higher at night, while in susceptible mice, the firing rate remained elevated in both the day and night leading to a blunted rhythm (S2C Fig). Since LHb cells exhibit silent, tonic, or burst-firing characteristics [12], we investigated whether exposure

to stress affected these diurnal firing characteristics in cells$^{LHb \to DRN}$. All 3 behavioural phenotypes exhibited a greater percentage of burst-firing cells at nighttime (**Fig 2H**). Intrinsic membrane properties also show diurnal rhythmicity such that during the day, cells$^{LHb \to DRN}$ of susceptible mice were significantly more excitable as evident by increased firing in response to current injection (**S3A -top Fig**). In accord with increased evoked excitability, susceptible mice displayed reduced threshold to induce the first spike (rheobase) in response to current injection and increased membrane input resistance in the steady-state current–voltage (I-V) relationship (**S3B and S3C -top Fig**). In contrast, at night, there was no difference in evoked excitability, rheobase, or input resistance between any of the 3 behavioural groups (**S3A, S3B, and S3C -bottom Fig**).

## Diurnal rhythms in $I_h$ and $K_v^+$-currents in cells$^{LHb \to DRN}$ of susceptible mice

Hyperpolarization-activated cation channel (HCN) mediate $I_h$-currents, which typically have an excitatory drive, undergo diurnal rhythmic expression, and are differentially regulated in susceptible and resilient mice [13,14]. We tested whether the diurnal differences in excitability in cells$^{LHb \to DRN}$ of susceptible and resilient mice was due to changes in the strength of the $I_h$-currents. Our observations that $I_h$-currents of control mice are equally low in the day and the night (**Fig 3A and 3B**) suggest that these channels do not drive the diurnal difference in firing in this circuit in nonstressed conditions. However, exposure to chronic stress leads to larger $I_h$-currents in the day than night in resilient and susceptible mice (**Fig 3A and 3B**). We used the HCN channel blocker ZD7288 to investigate the functional role of $I_h$-currents in driving spontaneous activity in cells$^{LHb \to DRN}$ of control, resilient, and susceptible mice. Bath application of ZD7288 excited some cells and inhibited others (**S4A Fig**). Moreover, a lack of difference in the percentage of excitation or inhibition on spontaneous firing between all groups implies that $I_h$-currents alone do not play a large role in increased firing in susceptible mice (**S4B and S4C Fig**).

Next, we explored the role of voltage-gated potassium ($K_v^+$)-channels since these channels are affected by stress-exposure and undergo diurnal rhythmic expression [14–16]. Specifically, we investigated whether these currents are responsible for the daily differences in firing in cells$^{LHb \to DRN}$. Since subtypes of $K_v^+$-channels exhibit different kinetics, we analyzed both peak (fast-kinetics) and sustained (slow-kinetics) $K_v^+$-currents in the day and night following CSDS. Interestingly, we found larger peak $K_v^+$-currents in the day in all behavioural groups with susceptible mice showing the largest peak current in the day than the night (**Fig 3C and 3D -top**). This indicates that fast-kinetics $K_v^+$-channels undergo daily rhythmic expression in this circuit, irrespective of stress-exposure. In contrast, sustained $K_v^+$-currents do not exhibit daily rhythmicity in control mice implying that slow-kinetics $K_v^+$-channels may not regulate diurnal changes in firing rate in cells$^{LHb \to DRN}$ in nonstressed conditions (**Fig 3C and 3D -bottom**). However, susceptible mice exhibit significantly larger sustained $K_v^+$-currents in the day compared to night (**Fig 3C -bottom**). It is likely that exposure to stress induces molecular changes that leads to rhythmic expression of slow-kinetic $K_v^+$-channels in susceptible, but not resilient mice in the daytime. In contrast, homeostatic adaptations in resilient mice may prevent the stress-induced diurnal expression of these channels. Since the majority of $K_v^+$-currents have an inhibitory drive, our observation that control and resilient mice express lower daytime spontaneous firing may in part be due to the daytime increase in peak $K_v^+$-currents. However, the large daytime peak and sustained $K_v^+$-currents in susceptible mice was unexpected. We next used a broad-spectrum $K_v^+$-channel blocker tetraethylammonium (TEA) to investigate the functional role of $K_v^+$-currents in driving spontaneous activity in cells$^{LHb \to DRN}$ of these mice. Bath application of TEA decreased spontaneous firing in a subset of cells, while

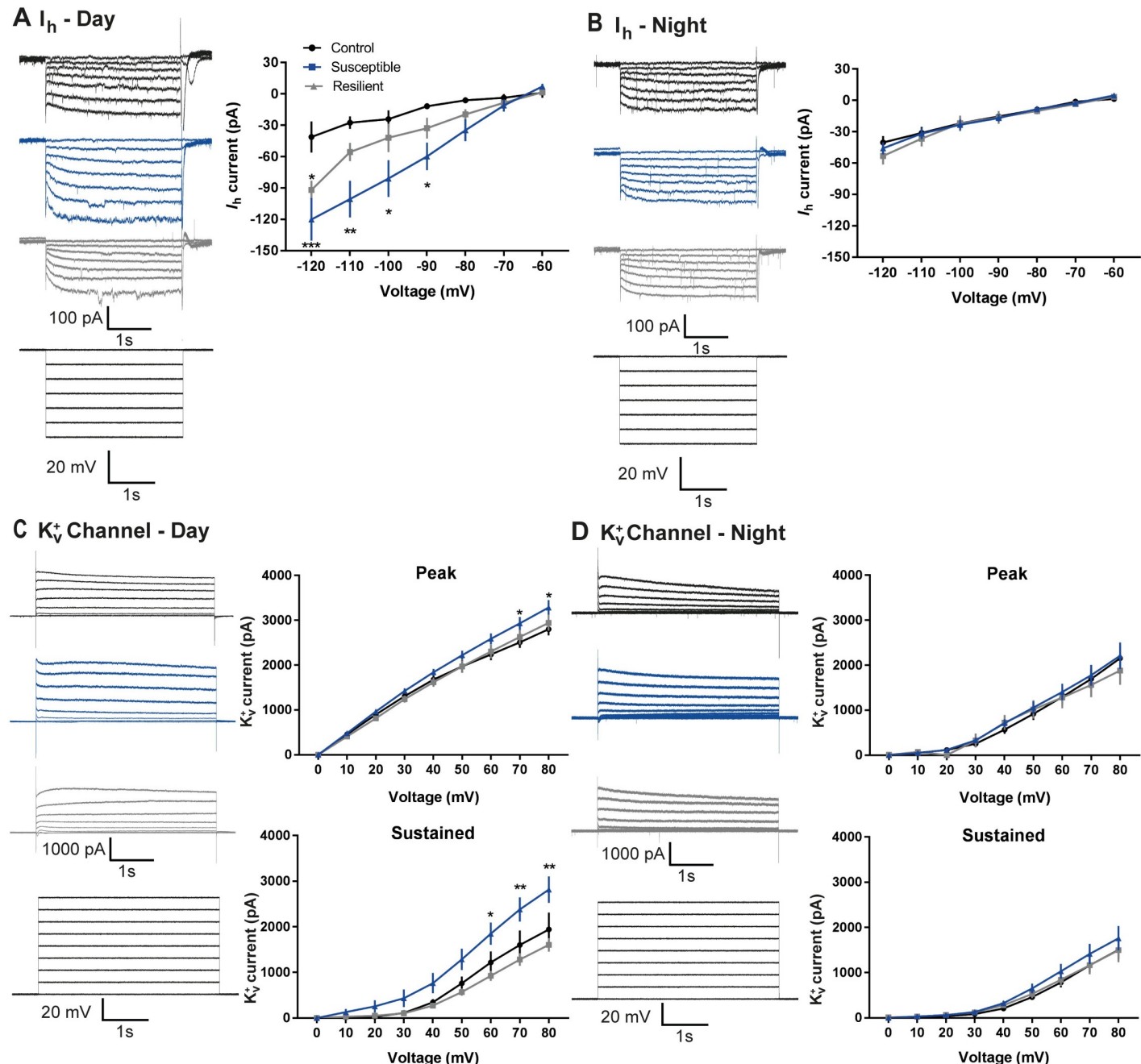

**Fig 3. Diurnal rhythms in $I_h$ and Kv+- currrents in cells$^{LHb \rightarrow DRN}$ of susceptible mice.** (**A**) Daytime representative traces (left) and statistical data of $I_h$-currents in response to incremental steps in voltage injections (right). Resilient and susceptible mice display significant increase in $I_h$-current compared to control ($F_{2,252}$ = 23.29, $P$ < 0.0001; $n$ = 10 to 14 cells from 7 to 11 mice per group). (**B**) Nighttime representative traces (left) and statistical data of $I_h$-currents in response to incremental steps in voltage injections (right). There was no difference in current between any of the phenotypes. (**C**) Daytime representative traces (left) and statistical data of isolated $K_v^+$ channel–mediated currents (right). Susceptible mice display significant increase in peak ($F_{2,216}$ = 9.94, $P$ < 0.0001; $n$ = 6 to 11 cells from 7 to 8 mice per group; top) and sustained ($F_{2,216}$ = 24.97, $P$ < 0.0001; $n$ = 6 to 11 cells from 7 to 8 mice per group; bottom) phases of $K_v^+$-currents. (**D**) Nighttime representative traces (left) and statistical data of isolated $K_v^+$ channel–mediated currents (right). There is no difference in peak and sustained phases of $K_v^+$-currents among the 3 phenotypes. Error bars: mean ± SEM. The raw data can be found in S3 Data.

in others, firing was increased (**S4D Fig**). TEA significantly attenuated daytime firing in susceptible mice compared to controls (**S4E Fig**). In contrast, there was no difference in the degree of inhibition between the groups at night. TEA-induced increased spontaneous firing was

generally higher in susceptible mice in both the day and night. The percentage excitation was significantly higher in susceptible mice compared to resilient groups in the night (**S4F Fig**).

## Daytime optical activation of cells$^{LHb \to DRN}$ during weak social stress induces susceptibility

Our observation that the spontaneous firing rate in cells$^{LHb \to DRN}$ of susceptible mice is elevated in the day and the night led us to speculate that blunted diurnal rhythmic firing in this circuit may represent pathophysiology leading to mood disorders. We performed in vivo optogenetics experiments to directly correlate the link between increased daytime firing in cells$^{LHb \to DRN}$ and stress-susceptibility. We first expressed channelrhodopsin (ChR2) in cells$^{LHb \to DRN}$ and then validated that optical stimulation with blue light (473 nm) induces action potentials and inward-currents in these cells (**S5A and S5B -left Fig**). Frequency-response curves of membrane excitability showed fidelity up to 20 Hz following optical stimulation (**S5B -right Fig**). Next, we performed in vivo optogenetic experiments where mice were exposed to a weak, 7-day social defeat stress (SDS) paradigm during which the ChR2 expressing cells$^{LHb \to DRN}$ were optically stimulated for 20 min after social stress (**Fig 4A**; **S5C and S5D Fig**). Studies suggest that mice that undergo SDS do not typically exhibit social avoidance or other stress-susceptible behaviours but are more vulnerable to subsequent stress [17,18]. Optical stimulation of cells$^{LHb \to DRN}$ during SDS (SDS-ChR2) induced stress-susceptibility as shown by the significant decrease in SI ratio compared to mice exposed to SDS alone (SDS-mCherry) or non-SDS-exposed mice that underwent optical stimulation of cells$^{LHb \to DRN}$ (non-SDS-ChR2: **Fig 4B**; **S5E Fig**). Our findings that pro-susceptibility was only observed when cells$^{LHb \to DRN}$ were optically stimulated during SDS suggests that the pro-stress effects of increased firing in this circuit is context specific. Moreover, the effect is long lasting because SDS-ChR2 mice, but not nonstressed SDS-mCherry and non-SDS-Chr2 groups, continue to show stress-susceptibility because they express anhedonic traits (decreased sucrose preference) even 36 days after the end of optogenetic stimulation during SDS (**Fig 4C**).

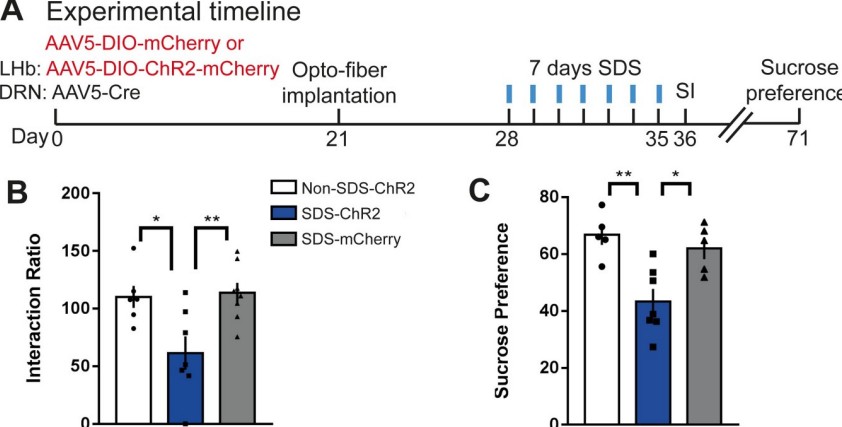

**Fig 4. Daytime optical activation of cells$^{LHb \to DRN}$ during weak social stress induces susceptibility. (A)** Experimental timeline of viral surgeries, optofiber implantation, repeated high frequency optical stimulation patterns, and behavioural tests. (**B**) In the presence of a CD1 social target (nonaggressor), SDS-ChR2 mice display increased stress-susceptibility as measured by decreased SI ratio compared with non-SDS-ChR2 and SDS-mCherry mice, ($F_{2,18} = 6.905$, $P < 0.01$; $N = 6$–8 mice/group). (**C**) The SDS-ChR2 mice display anhedonia as measured by reduction in 1% sucrose intake over 3 h in the sucrose preference test ($F_{2,14} = 9.805$, $P < 0.01$; $N = 5$–7 mice/group; right). Error bars: mean ± SEM. The raw data can be found in S4 Data. DRN, dorsal raphe nucleus; LHb, lateral habenula; SDS, social defeat stress; SI, social interaction.

## Susceptible or resilient mice exhibit different rates of photoentrainment

To assess the effects of a chronic social stressor on diurnal and circadian rhythms, we measured locomotor wheel-running activity in stress-naïve (control), susceptible, and resilient mice for 7 days during a standard 12 h:12 h LD cycle followed by 7 days of complete darkness (DD) (Fig 5A; S6A Fig). Exposure to CSDS did not affect endogenous rhythms since there was no difference in the phase of activity onset in DD relative to LD between any of the groups (S6B–S6D Fig). Moreover, there was no difference in the free-running period in DD and amount of activity in LD and DD (S6E and S6F Fig). Further detailed analysis of behavioural rhythms showed that intradaily variability was lower in stress-exposed mice compared to stress-naïve groups, while there was no difference for interdaily stability between the 3 phenotypes (S6G and S6J Fig). We also measured photoentrainment in stress-exposed mice challenged by a 6-h advance in LD cycle. Behavioural wheel running activity was measured for 7 days in standard LD cycle followed by activity measurement for 7 days after a single 6-h advance in light onset (Fig 5A and 5B). Susceptible mice exhibited significantly slower initial rate of photoentrainment, while resilient mice were significantly faster at fully reentraining to the new photoperiod (Fig 5C and 5D).

## Blunted rhythms in spontaneous firing in DRN cells of susceptible mice

We also investigated spontaneous firing in unlabeled DRN cells during the day and the night (Fig 6A). Daytime spontaneous firing in DRN cells is lower in susceptible, and higher in

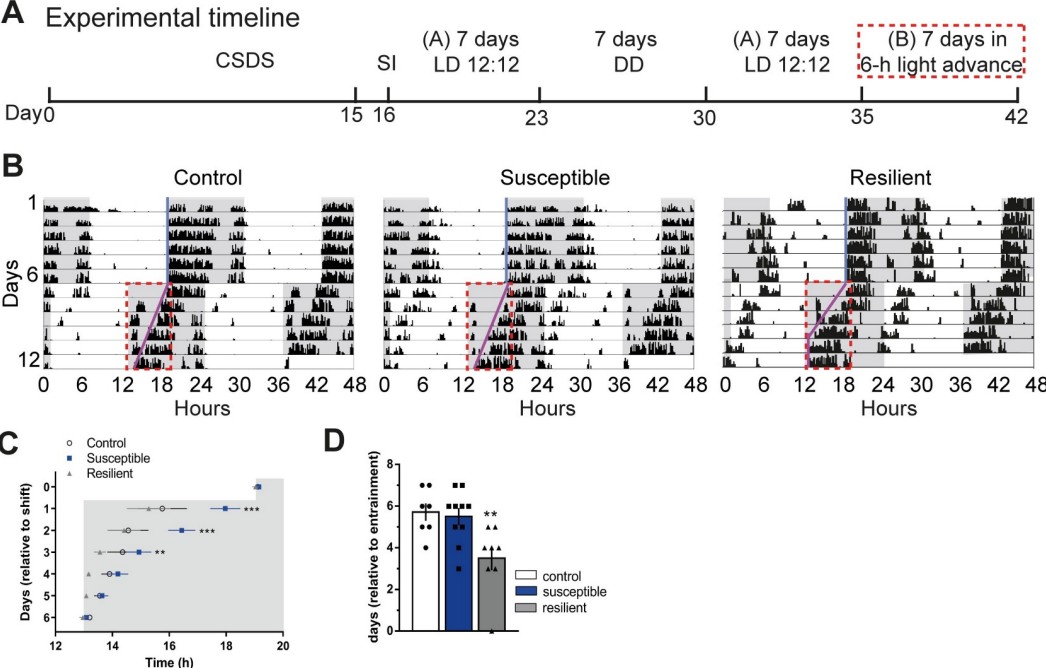

**Fig 5. Susceptible or resilient mice exhibit different rates of photoentrainment. (A)** Experimental timeline (A, normal light cycle: Lights ON: 07:00; Lights OFF: 19:00; B, new light cycle following 6 h advance: Lights ON: 13:00; Lights OFF: 01:00), red box denotes 6 h lights advance in new LD cycle. **(B)** Representative double-plotted actograms of activity before and after 6 h shift in LD cycle. Grey background indicating dark phase of the LD cycle. Light-blue and pink lines representing extended regression line derived by onset of activity under normal and new LD cycles, respectively. Dotted red lines outlining the data of activity onsets shown in (c). **(C)** Susceptible mice display significantly slower initial rate of photoentrainment in the new LD cycle ($F_{2,25} = 10.81$, $P < 0.001$) **(D)** Resilient mice took significantly less number of days to reentrained in the new LD cycle ($F_{2,22} = 6.587$, $P < 0.05$). $N = 7–10$ mice/group. Error bars: mean ± SEM. The raw data can be found in S5 Data. CSDS, chronic social defeat stress; DD, complete darkness; LD, light–dark; SI, social interaction.

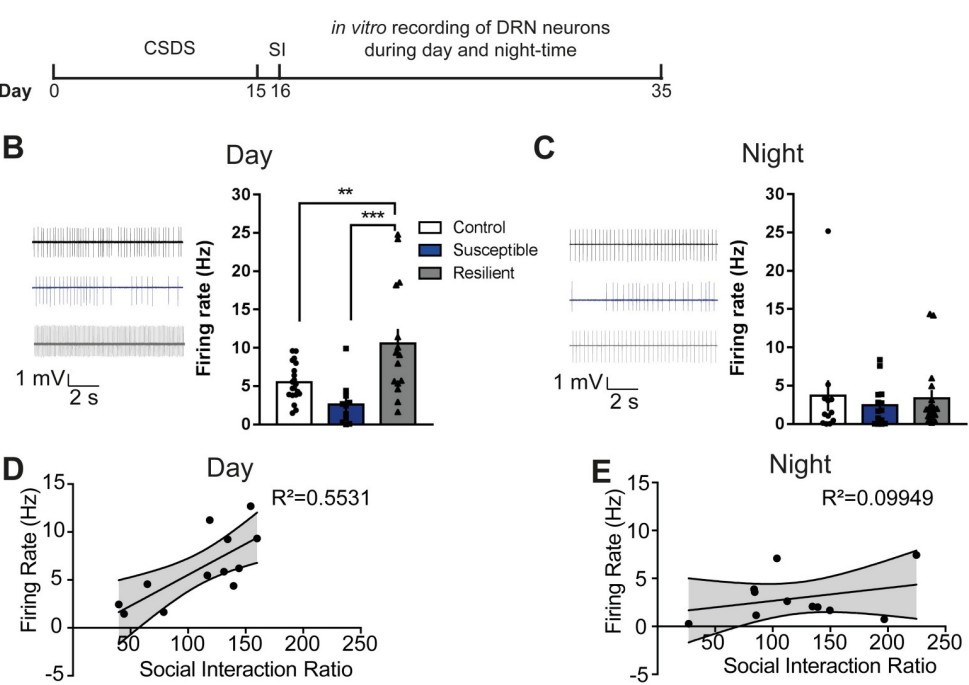

**Fig 6. Blunted rhythms in spontaneous firing in DRN cells of susceptible mice. (A)** Experimental timeline of CSDS paradigm, SI test, and in vitro recordings. **(B, C)** Sample traces of spontaneous firing of DRN cells in day (B) and in night (C) in control (top), susceptible (middle), and resilient mice (bottom). **(B)** Susceptible mice exhibit decreased daytime spontaneous firing in DRN cells, while resilient mice exhibit increased daytime spontaneous firing ($F_{2,43}$ = 9.793, $P < 0.001$; $n = 11$–$20$ cells from 7 to 11 mice/group), **(C)** No difference in nighttime spontaneous firing in DRN cells. **(D, E)** Significant positive correlation between spontaneous activity in DRN cells and SI ratio during the day ($R^2$ = 0.5531; $P < 0.05$) but not the night. Error bars: mean ± SEM. The raw data can be found in S6 Data. CSDS, chronic social defeat stress; DRN, dorsal raphe nucleus; SI, social interaction.

resilient mice, compared to control groups, while at night, there was no difference between the groups (**Fig 6B and 6C**). Immunohistochemical analysis shows that synaptic terminals from LHb projection to the DRN overlap with both putative DRN GABAergic interneurons and putative DRN 5HT cells (**S7A–S7D Fig**). Furthermore, there was a significant positive correlation between firing rate and the SI ratio during the day but not at night (**Fig 6D and 6E**). Comparison of diurnal rhythms in firing shows that control and resilient mice tend to express higher relative activity in the day, while susceptible mice show blunted rhythms where activity is equally low in the day and the night (**S7B Fig**).

## Discussion

We found that cells$^{LHb \rightarrow DRN}$ of susceptible, but not resilient mice, exhibit pathophysiological firing such that activity is elevated in both the day and the night. However, it should be noted that although we labeled cells which project from the LHb to DRN, it is possible that these same LHb neurons also project to sites distinct from the DRN, that may be responsible for the effects on mood observed in our study. The LHb and DRN have direct and indirect connections with the SCN which likely leads to their functional role in integrating mood related behaviour with circadian timing [1,4,19]. Thus, we predict that stress-induced pathophysiological firing in the putative loop consisting of the LHb, DRN, and SCN (LHb→DRN→SCN→LHb) may be responsible for changes in daily rhythms associated with

mood disorders. Stress and the circadian system exhibit bidirectional interaction where, for example, clock proteins CRY 1/2 increase the responsiveness of the HPA-axis [20], while stress-induced increase in glucocorticoids leads to altered clock gene expression in the SCN and regions associated with mood regulation [21,22]. Stress increases firing in the LHb which activates the HPA-axis causing elevated corticosterone release [23]. Thus, our observations of elevated diurnal firing in cells$^{LHb \rightarrow DRN}$ of susceptible mice may be responsible for blunted rhythms in corticosterone, observed in rats exposed to chronic mild stress [24]. Blunted diurnal rhythmic expression of clock proteins in the SCN of stress-susceptible rodents [25,26] likely causes blunted rhythmic firing of SCN cells projecting to downstream targets such as the LHb. We speculate that this process may be responsible for the elevated spontaneous firing in cells$^{LHb \rightarrow DRN}$ during the day and the night in susceptible mice, leading to blunted diurnal rhythmicity in neural activity. In contrast, homeostatic adaptations in the SCN and LHb cells of resilient mice may prevent stress-induced blunting of rhythmic activity. Moreover, the blunted rhythm is inverted in susceptible mice, while low daytime and high nighttime firing in control and resilient mice correlates with previous observations [8,9]. The inversion of diurnal firing in susceptible mice may be due to stress induced shifts in clock gene expression in LHb cells resulting in changes in ion channel expression and neural firing. Elevated diurnal expression of clock gene *Per 2* in the LHb following stress [22] may be a possible mechanism driving abnormal firing in susceptible mice. Abnormal levels of stress hormone may be another mechanism where high concentrations increase neural firing while prolonged exposure to high concentrations decreases neural firing [27,28]. Since stress-exposed mice exhibit prolonged diurnal levels of stress hormones [29], abnormally elevated levels of daytime stress hormones in susceptible mice may activate pathways that increase firing, while prolonged concentrations into the night decreases relative firing. In contrast, homeostatic adaptations in resilient mice may lead to low daytime hormone levels that leads to low firing, while the relative increase in nighttime stress hormone levels leads to increased firing.

Studies demonstrate that synaptic inputs and diurnal changes in ion channel expression regulate neural excitability, control membrane potential, and duration of action potential [30]. Therefore, we speculated that stress-induced disruptions on the daily oscillations of hyperpolarization-activated cation channel (HCN)-mediated $I_h$-currents and voltage-gated potassium ($K_v^+$)-currents in cells$^{LHb \rightarrow DRN}$ may be responsible for the elevated diurnal, spontaneous firing in susceptible mice. It is well established that nighttime excitability in paraventricular neurons is driven by large $I_h$-currents, while low firing correlates with low $I_h$-currents [31]. Thus, we predicted that low $I_h$ and high $K_v^+$-currents during the day in control and resilient mice drive low firing, while high $I_h$ and low $K_v^+$-currents drive high nighttime firing. In contrast, elevated $I_h$- and decreased $K^+$-currents at both the day and the night would drive increased diurnal firing in susceptible mice. Our findings that both $I_h$- and sustained $K_v^+$-currents are equally low in the day and night of control mice suggest that these currents do not have a large functional role in regulating daily rhythms in spontaneous firing in cells$^{LHb \rightarrow DRN}$ in nonstressed conditions. However, susceptible mice show rhythmic expression of these currents (high at day, low at night) possibly due to stress-induced changes in molecular pathways triggering increased expression of $I_h$ and slow kinetic $K_v^+$-channels in the day. Spontaneous firing is a net measure of synaptic inputs and the summed effect of numerous types of ion channels. Therefore, this may account for the discrepancy between robust daily rhythms in $I_h$- and $K_v^+$-currents but not in spontaneous firing in susceptible mice. Somatic HCN channels, which typically increase cell firing, likely do not play a significant role in driving excitability in cells$^{LHb \rightarrow DRN}$ in susceptible mice since HCN antagonist ZD7288 equally attenuated firing in the day and the night in all groups. The excitatory effect of the antagonist may be due to block of dendritic HCN2 channels that increases local dendritic membrane time constant that delays the decay of colocalized

AMPA receptor-induced excitatory postsynaptic potential (EPSP), resulting in the temporal summation of EPSP and increased probability of firing [32,33]. The bidirectional effects of ZD7288 on spontaneous firing implies the existence of at least 2 subpopulations of cells$^{LHb \rightarrow DRN}$ with differential expression of HCN channel subtypes.

The increased number of cells exhibiting bursting activity at night in all the behavioural phenotypes may reflect an increased expression of T-type calcium ($Ca^{2+}$)-channels and calcium-activated-$K^+$- (BK) channels which have been previously shown to regulate bursting activity in the LHb [34]. Since $K_v^+$-currents typically have an inhibitory drive that decreases the firing rate of a cell, the daytime increased peak voltage activated ($K_v^+$)-currents in cells$^{LHb \rightarrow DRN}$ in control and resilient mice may be partly responsible for the lower activity in these mice. However, a large peak and sustained $K_v^+$-currents in cells$^{LHb \rightarrow DRN}$ in susceptible mice is unexpected. How can large $K_v^+$-currents account for elevated firing? Two possible explanations are: (i) increased $I_h$-currents induce compensatory $K_v^+$-currents [14]; however, these compensatory currents in cells$^{LHb \rightarrow DRN}$ may be ineffective at lowering the firing rate possibly due to a larger net increase in the excitatory driving force; and (ii) aberrations in clock genes expression up-regulate daytime expression of fast activation and inactivation A-type $K_v^+$-channels (IKA) channels and delayed rectifier $K_v^+$-channels (DR). These channels increase firing rate by shortening action potential duration. Moreover, these channels undergo daily oscillations within the SCN where peak expression is highest in the day when these cells have the highest firing rate [15,16]. Thus, diurnal rhythms in peak $K_v^+$-currents in cells$^{LHb \rightarrow DRN}$ may represent rhythmic expression of the fast dynamics IKA-channels whose daytime expression is further increased in susceptible mice. Furthermore, large daytime-sustained $K_v^+$-currents in susceptible mice may represent a larger daytime expression of DR channel that returns to basal levels at night. Observations that the $K_v^+$-channel antagonist TEA, which blocks DR and BK channels, significantly attenuated daytime firing in cells$^{LHb \rightarrow DRN}$ of susceptible mice correlates with the higher daytime $K_v^+$-currents in susceptible mice. Elevated expression of putative DR and/or BK channels may drive increased daytime activity in cells$^{LHb \rightarrow DRN}$ of susceptible mice. Moreover, corticotrophin-releasing factor (CRF), which is elevated in susceptible mice, increases firing in the LHb cells via BK-channel activation [35,36] and further implicates a role for elevated BK-channels driving daytime firing in susceptible mice. The increased inhibition of spontaneous firing by TEA in resilient mice is less clear as we did not observe large daytime $K_v^+$-currents in this phenotype. The excitatory effects of TEA on a subset of cells is likely due to nonspecific antagonism of $K_V^+$-channels that functionally decrease cell firing. In both the day and night, the percentage of excitation was significantly higher in susceptible mice compared to control and resilient groups which is likely indicative of larger expression of $K_V^+$-channels that typically suppress spontaneous firing. Since nighttime $K_V^+$-currents was similar in all phenotypes, the greater excitatory effect with TEA in susceptible mice is unexpected. One possible explanation may be that removal of inhibitory tone increases excitatory tone to a greater degree in these cells. Future studies will elucidate the potential role of the subtypes of $K_V^+$-channels in driving diurnal excitability in cells$^{LHb \rightarrow DRN}$ of stress-exposed mice.

The differences in the rates of reentrainment to a new LD cycle between susceptible and resilient mice suggest that stress induces different changes in molecular and cellular processes that regulate photoentrainment. Phosphorylated glycogen synthase kinase (p-GSK-3β) [37]is decreased in susceptible mice and since photoentrainment requires conversion of p-GSK-3β to GSK3β [38], aberrations in this cascade may be responsible for the initial slow rate of photoentrainment observed in the susceptible phenotype. Salt-inducible kinase 1 (SIK1) in the SCN is part of a negative feedback loop that prevents immediate photoentrainment to nighttime light pulses [39]. Since stress increases SIK1 expression [40], elevated light-induced SIK1 expression in susceptible mice may be responsible for the slower initial rate of photoentrainment,

while homeostatic adaptations in resilient mice may lead to reduced light-induced SIK1 expression leading to faster photoentrainment. Independent oscillations of *Per* expression in the LHb highlights autonomous clock function in this nucleus that may drive certain circadian behaviours [3]. Moreover, the SCN output signaling molecule prokineticin 2 (PK2) conveys time of day information to the rest of the brain, including the LHb where it decreased neural firing by increasing GABAergic tone [41,42]. Thus, elevated day and night firing, together with delayed initial rate of photoentrainment, in susceptible mice be indicative of decreased, and blunted rhythmic, PK2 signaling from the SCN. Observations that light pulses increases firing in a subset of cells in the LHb [42] together with reports that hyperactivation of LHb cells disturbs sleep, possibly via activation of DRN cells [43], highlight another putative mechanisms by which elevated firing in cells$^{DRN \to SCN}$ slows the initial rate of photoentrainment in susceptible mice.

Increased activity of excitatory cortical projections to the DRN of susceptible mice decreases firing in DRN cells via feedforward inhibition [11]. Thus, the decreased diurnal firing in DRN cells of susceptible mice may be indicative of robust feedforward inhibition due to hyperactive cells$^{LHb \to DRN}$ in both the day and the night. In contrast, high daytime firing in the DRN of control and resilient mice may be indicative of low firing in cells$^{LHb \to DRN}$, while low nighttime firing in the DRN may be indicative of higher firing in cells$^{LHb \to DRN}$. Moreover, immunohistochemical observation that LHb terminals in the DRN surround both putative GABAergic interneurons and 5HT cells implies that glutamatergic LHb terminals make direct (excitatory) synaptic contact with 5HT cells or indirect synaptic contact via feedforward inhibition involving GABAergic interneurons. DRN cells including those that have direct (cells$^{DRN \to SCN}$) and indirect projections to the SCN (cells$^{DRN \to \to SCN}$) also regulate nonphotic entrainment [44,45]. Nonphotic cues such as temperature arousal and sleep–wake patterns are affected in susceptible mice [46–48]. Moreover, the DRN has a functional role in regulating body temperature, sleep–wake cycle, and arousal [1,49,50]. Thus, pathophysiological firing in cells$^{LHb \to DRN}$ and DRN nuclei of susceptible mice may also affect nonphotic entrainment resulting in slower initial rate of reentrainment in susceptible mice. However, the modulatory role of 5-HT on photic responses in the SCN together with recent observations that decreased VTA dopamine input to the SCN slows the rate of photoentrainment [51] leads us to hypothesize that decreased diurnal firing in putative cells$^{DRN \to SCN}$ of susceptible mice may be responsible for the delayed initial rate of photoentrainment. Moreover, faster photoentrainment in resilient mice may be indicative of homeostatic adaptations resulting in large increases in daytime firing observed in DRN cells, a subset of which may be projecting to the SCN. These putative cells$^{DRN \to SCN}$ may modulate SCN cells such that the clock system is able to rapidly entrain to the new photostimulus.

This is the first demonstration, to our knowledge, that susceptible mice exhibit: (i) pathophysiological changes in cells$^{LHb \to DRN}$ that lead to elevated diurnal firing, leading to blunted and inverted rhythmic activity; (ii) decreased diurnal firing in the DRN, leading to blunted rhythmic activity; and (iii) slower initial rate of photoentrainment. We speculate that susceptible mice may express dysfunctional cellular homeostatic mechanisms that lead to abnormal circadian clock gene rhythms that alters firing in circuits that regulate mood and photoentrainment. In contrast, homeostatic adaptive changes in resilient mice may buffer against pathophysiological changes. These studies continue to deepen our understanding of the link between stress and biological rhythms.

## Materials and methods

### Animals

All experiments performed were approved by the NYUAD Animal Care and Use Committee, and all experimental protocols were conducted according to the National Institute of Health

Guide for Care and Use of Laboratory Animals (IACUC Protocol: 150005A2). CD1 retired male breeders (Charles River) and C57BL/6J male mice (8 to 12 weeks; Jackson Laboratories) were used in all experiments of this study. All mice were maintained in the home cages, with ad libitum access (unless noted otherwise) to food and water in temperature (23 ± 2˚C)- and humidity (50 ± 10%)-controlled facilities with 12 h light–dark (L/D) cycles (lights on: 7:00 AM; lights off: 7:00 PM, Zeitgeber time ZT; ZT0—lights on; ZT12—lights off). In phase-shift experiments, mice were first entrained to the standard LD cycle after which they were exposed to a 6-h phase advance where lights came on at 1:00 AM and lights off at 1:00 PM. All behavioural tests were conducted during the light cycle (ZT 5 to 10), and mice were habituated to the recording room and lighting conditions for at least 1 h. Between trials, the behavioural apparatus was cleaned with MB-10 solution (Quip Laboratories, United States of America) to avoid persistence of olfactory cues.

## Viral vectors

AAV5-EF1a-DIO-mCherry and AAV5-EF1a-DIO-hChR2 (H134R)-mCherry-WPRE-pA virus plasmids were purchased from the University of North Carolina vector core facility (UNC). Retrograde AAV5-CMV-PI-Cre-rBG virus was purchased from the University of Pennsylvania viral vector core facility (UPenn).

## Stereotaxic surgery, viral mediated gene transfer, and optic fiber placement

Mice were anaesthetized with a ketamine (100 mg kg$^{-1}$) and xylaxine (10 mg kg$^{-1}$) mixture, placed in a stereotaxic apparatus (RWD Life Science), and their skull was exposed by scalpel incision. For injection of replication defective viral vectors, 33-gauge needles were used. For injection of Cre-dependent AAV5-EF1a-DIO-mCherry virus, needles were placed bilaterally at a 0˚ angle into the LHb (in mm: anterior/posterior, −1.6; lateral/medial, 0.4; dorsal/ventral, −3.0), and 0.2 μl of virus was infused at a rate of 0.1 μl min$^{-1}$. AAV5-EF1a-DIO-hChR2 (H134R)-mCherry-WPRE-pA virus was injected bilaterally into the LHb at a 15˚ angle (in mm: anterior/posterior, −1.6; lateral/medial,1.1; dorsal/ventral, −3.1), and 0.2 μl of virus was infused at a rate of 0.1 μl min$^{-1}$. For retrograde travelling AAV5-CMV-PI-Cre-rBG viral injection, the needle was placed unilaterally at a 20˚ angle into the DRN (in mm: anterior/posterior, −4.7; lateral/medial, 1.1; dorsal/ventral, −3.2), and 0.5 μl of virus was infused at a rate of 0.1 μl min$^{-1}$. The needles were left in place for 10 min following injections to minimize diffusion and then completely withdrawn after viral delivery. For in vivo optical control of LHb neuronal firing in cells expressing hChR2-mCherry or mCherry, we used the chronically implantable optical fiber system. Three weeks after surgery, chronically implantable homemade fibers (200 μm core optic fiber) were implanted above the LHb at a 15˚ angle (anterior–posterior, −1.6 mm; lateral–medial, 1.1 mm; dorsal–ventral, −3.0 mm). In order to ensure secure fixture of the implantable fiber, the skull was dried and then industrial-strength dental cement (Zinc polycarboxylate cement; Factory of oral medicine materials at Stomatology School of Wuhan University) was added between the base of the implantable fiber and the skull. Mice were allowed to recover for at least 7 days before starting the behavioural paradigm.

## Blue light stimulation

Optical fibers (Thor Labs, BFL37-200, New Jersey, USA) were connected using an FC/PC adaptor to a 473 nm blue laser diode (SLOC Lasers, BL473T8-150FC, Shanghai, China), and a stimulator (Agilent Technologies, no. 33220A, California, USA) was used to generate blue light pulses. For in vitro slice electrophysiological validation of ChR2 activation, we tested 0.1 to 50 Hz (20 ms) stimulation protocols delivered to LHb neurons expressing ChR2 through an

optic fiber (Thor Labs, BFL37-105) attached to a 473-nm laser. For all in vivo behavioural experiments, we injected mice with a cre-dependent AAV5-EF1a-DIO-mCherry or AAV5-EF1a-DIO-hChR2(H134R)-mCherry-WPRE-pA into LHb and replication defective retrograde travelling AAV5-CMV-PI-Cre-rBG into the DRN to specifically label the LHb→DRN circuit, and chronically implantable homemade fibers with 200 mm core optic fiber were implanted into the LHb as described above. All mice were handled for a minimum of 1 min per day for 4 days. Three days before the experiment, "dummy" optical patch cables were connected to the fibers implanted in mice each day for 10 min in order to habituate them to the tethering procedure. During 7 days of subthreshold social defeat, mice received bilateral stimulation immediately after 10 min of physical interaction: Mice were given >1 mW laser with a stimulation frequency of 20 Hz and a 20-ms width light pulse (500 ms optical stimulation with an interstimulus interval of 1 s) for 20 min/day during sensory contact.

## Chronic social defeat stress (CSDS) paradigm and social interaction (SI) test

Both CSDS paradigm or subthreshold social defeat stress (SSDS) paradigm were performed according to previously published protocols [14,17,18,52,53] with minor modifications in our lab. Briefly, CD1 aggressor mice were housed in social defeat cages 24 h before the onset of defeats on one side of a clear perforated plexiglass divider. The experimental C57 mice were individually exposed to an aggressive CD1 mouse for 10 min during which time they were physically attacked by the CD1 mouse. After 10 min of physical contact, the mice were separated by the clear perforated plexiglass divider. For the following 24 h, the aggressor and experimental mouse were maintained in sensory contact using the perforated plexiglass partition dividing the resident home cage in two. To avoid habituation, the experimental mice were exposed to a new CD1 aggressor mouse home cage each day for 15 consecutive days. The control mice were housed in pairs within a cage continuously separated by a clear perforated plexiglass divider. On the day following the last day of defeat, the mice were singly housed in new cages. SI tests were performed on day 16. For the SSDS paradigm, the procedure was identical to the standard CSDS paradigm, with the exception that the procedure lasted for 7 consecutive days. In order to stimulate LHb cells projecting to DRN, 200 mm core optic fibers were attached to the chronically implanted fibers, after which the experimental mice expressing hChR2-mCherry or mCherry in LHb-DRN circuit underwent bilateral blue light stimulation (20 min/day per hemisphere) during the sensory-stress period, immediately after 10 min of physical contact. The standard social defeat paradigm was carried out between 16:00 and 17:00, and subthreshold social defeat were performed between 13:00 and 17:00. Social avoidance behaviour towards a novel CD1 nonaggressive mouse was measured in a two-stage SI test. In the first 2.5 min test (CD1 social target absent in the interaction zone), the experimental mouse was allowed to freely explore a square-shaped arena (44 × 44 cm) containing Plexiglas perforated cage with a wire mesh (10 × 6 cm) placed on one side of the arena. In the second 2.5-min test, the experimental mouse was reintroduced back into the arena with an unfamiliar CD1 nonaggressive mouse contained behind a Plexiglas perforated cage. TopScan video tracking system (CleverSys) was used to automatically monitor and record the amount of time the experimental mouse spent in the "interaction zone" (14 × 26 cm), "corner zone" (10 × 10 cm), and "total travel" within the arena for the duration of the 2.5-min test session (in absence and presence of the social target). Interaction zone time, corner zone time, total distance travelled, and velocity were collected and analyzed. The segregation of susceptible and resilient mice was based on the SI ratio, which was calculated as [100 × (time spent in the interaction zone during social target present session) / (time spent in the interaction zone during no social target

session)] as described previously. All mice with scores 100 were classified as "susceptible," and those with scores ≥100 were classified as "resilient." Mice that were used for either daytime or nighttime electrophysiology recordings all underwent CSDS between ZT9 and 10, while the SI tests were performed between ZT 7 and 9.

## Sucrose preference test

To measure anhedonia, sucrose preference test was performed as we previously reported [34,54,55]. After completion of the SI test, animals were single housed and habituated to 2 bottles of 1% sucrose for 2 days, followed by 24 h water and food deprivation. In the 3-h test period, the animals were exposed to 1 bottle of 1% sucrose and 1 bottle of water, with bottle positions switched half way through the experiment to ensure that the mice did not develop a side preference. The sucrose and water bottles were weighed before and after the test, recording the total consumption of each liquid. Sucrose preference was calculated as a percentage [100 × amount of sucrose consumption/total amount consumption (water + sucrose)].

## Circadian rhythms assay

Circadian activity rhythms of mice were determined in individual cages (33.2 × 15 × 13 cm) equipped with stainless steel wheels (11 cm inside diameter, 5.4 cm wide; Model ACTI-PT2-MCR2, Actimetrics, Illinois, USA). Wheel-running cages were placed in circadian cabinets (Phenome Technologies, Illinois, USA) with 6 cages per row. The light and temperature of the chambers was controlled by the ClockLab Chamber Control software (ACT-500, Actimetrics, Illinois, USA). Circadian activity rhythms were measured with infrared (clickless) sensor clips onto the lip and rail of the cage that detected wheel rotations, and the number of wheel revolutions every minute. Wheel rotation from the sensor was transmitted via a single channel connected to the ClockLab digital interface (ACT-556, Actimetrics, Illinois, USA) and recorded by a personal computer. Data were collected and analyzed using ClockLab Data Collection software (ACT-500, Actimetrics, Illinois, USA). All data was analyzed by a trained scorer blind to treatment. To record the circadian activity rhythms, mice were individually housed in activity wheel-equipped cages (Actimetrics, Illinois, USA) in light-tight boxes under a 12-h LD cycle. (light intensity 180 to 200 lux; temperature inside light-tight boxes: 25.5 ± 1.5°C) for at least 7 days, then transferred to constant and complete darkness (DD) cycle and provided with food and water ad libitum. Wheel running rhythms were monitored and analyzed with ClockLab Data Collection system (ACT-500, Actimetrics, Illinois, USA). The free-running period was calculated according to the onset of activity across 7 days in constant darkness. Onset of activity was identified as the first bin above a threshold of 5 counts preceded by at least 6 h of inactivity and followed by at least 6 h of activity and offsets were determined by at least 6 h of activity and followed by 6 h of inactivity through ClockLab software. Onset and offset points were edited by eye when necessary. A regression line was fit through activity onsets for the 7 days of LD cycle and extrapolated for 7 days following DD cycle. The magnitude of phase shifts was calculated as the time difference between the 2 lines of LD and DD cycle. To assess reentrainment to a 6-h advance to the LD cycle, mice were kept in a 12-h LD cycle (light intensity 180 to 200 lux; temperature inside light-tight boxes: 25.5 ± 1.5°C) for at least 7 days. Following this entrainment, the onset of darkness of the cycle was abruptly advanced 6 h and daily advances (hours) in running wheel activity onsets were recorded every day for an additional 7 days. The days required to entrain following the shift in the light cycle was calculated. The duration of reentrainment was defined as the number of days required to shift activity onset by 6 ± 0.25 h followed by 2 consecutive days of activity onset within this range.

## In vitro patch-clamp electrophysiology

Cell-attached or whole-cell recordings were obtained from LHb neurons projecting to DRN in acute brain slices from C57BL/6J mice that had been stereotaxically injected with AAV5-EF1a-DIO-mCherry or AAV5-EF1a-DIO-hChR2(H134R)-mCherry-WPRE-pA into the LHb and retrograde viral vector AAV5-CMV-PI-Cre-rBG in DRN. To minimize stress and to obtain healthy LHb slices, mice were anaesthetized with isofluorane and perfused immediately for 40 to 60 s with ice-cold artificial cerebrospinal fluid (aCSF), which contained (in mM): 128 NaCl, 3 KCl, 1.25 NaH2PO4, 10 D-glucose, 24 NaHCO3, 2 CaCl2, and 2 MgCl2 (oxygenated with 95% O2 and 5% CO2 (pH 7.4), 295 to 305 mOsm). Acute brain slices containing LHb neurons were cut using a microslicer (DTK-1000, Ted Pella) in ice-cold sucrose aCSF, which was derived by fully replacing NaCl with 254 mM sucrose and saturated by 95% O2 and 5% CO2. Slices were maintained in holding chambers with aCSF for 1 h recovery at 37˚C and then at room temperature for recording. Cell-attached and whole-cell recordings were carried out using aCSF at 34˚C (flow rate = 2.5 ml min$^{-1}$). Patch pipettes (5 to 8 MΩ for cell-attached recordings, and 3 to 5 MΩ for whole-cell recordings) were filled with internal solution containing the following (in mM): 115 potassium gluconate, 20 KCl, 1.5 MgCl2, 10 phosphocreatine, 10 HEPES, 2 magnesium ATP, and 0.5 GTP (pH 7.2, 285 mOsm). For measurements of the spontaneous activity of mCherry-labeled LHb-DRN neurons, cell-attached recordings were performed in acutely prepared LHb-containing brain slices. Signals were band-pass filtered at 300 Hz to 1k Hz and then Bessel filtered at 10 kHz (gain 1) using a Multiclamp 700B amplifier. To measure the intrinsic membrane properties of LHb neurons, whole-cell recordings were carried out in current-clamp mode, and spikes were induced by incremental increases of current injection (each step increase was 25 pA; range −10 to 200 pA). Whole-cell voltage clamp was used to record Ih-current with a series of 2 s pulses with a 10-mV command voltage step from −120 mV to −60 mV from a holding potential of −60 mV. To isolate voltage-gated K$^+$ channel-mediated currents, 3 s pulses with a 10-mV from 0 mV to +80 mV from a holding potential of −60 mV.

For whole-cell recordings during optogenetic stimulation, resting membrane potential and action potentials were recorded in current-clamp mode and inward current measurements were made in voltage-clamp mode using the Multiclamp 700B amplifier and data acquisition was collected using a Digidata 1550B digitizer and Clampfit 10.5 (Molecular Devices, California, USA). Series resistance (15 to 25 mΩ) was monitored during the experiments and membrane currents and voltages were filtered at 10 kHz (Bessel filter), and cells were discarded from further analysis if, under their basal conditions, this value changed by more than 15%. For in vitro electrophysiological validation of ChR2 activation, whole-cell optogenetic recordings were obtained from LHb neurons in acute brain slices from C57BL/6J mice that had been stereotaxically injected with AAV5-EF1a-DIO-hChR2(H134R)-mCherry-WPRE-pA into the LHb and retrograde viral vector AAV5-CMV-PI-Cre-rBG in DRN. Sustained and trains (0.1 to 50 Hz) of blue light were generated by a stimulator (described above) and delivered to LHb neurons expressing ChR2 through a 105-mm optic fiber attached to a 473-nm laser. Daytime recordings were performed at ZT2 to 7, while nighttime recordings were performed at ZT14 to 19. Cell-attached recordings we performed to measure the effects of HCN antagonist (ZD7288 10 μM) or K$^+$-channel blocker TEA (10 mM) on spontaneous firing in cells$^{LHb→DRN}$. Baseline spontaneous firing were recorded for 3 min after which the antagonist was perfused unto the slice chamber for 9 min after which spontaneous firing was recorded again for 3 min. Access resistance was monitored throughout, and cells were discarded from further analysis if resistance changed by 40%. All daytime electrophysiology recordings were carried out between ZT 2 and 7, and all nighttime electrophysiology recordings were carried out between ZT 14 and 19. To

calculate the effect of drug on spontaneous firing, the following equation was used: (frequency of spontaneous firing after drug / frequency of spontaneous firing before drug)*100.

## In vivo extracellular single-unit electrophysiology

Mice were anesthetized with 10% chloral hydrate (400 mg/Kg) and head fixed horizontally onto a stereotaxic frame. Using bregma, LHb was located within the range (in mm): anterior/posterior: −0.94 to −2.18, medial/lateral: 0.25 to 0.80, dorsal/ventral: −2.50 to −2.80. Glass micropipettes (15 to 20 MΩ) filled with 2M NaCl was used for recording. Recorded electrical signals were amplified and band-pass filtered (0.3k to 1k Hz) using a DP-311 Differential Amplifier (Warner Instruments, Massachusetts, USA). Data acquisition of in vivo spontaneous firing activity was collected using a Digidata 1550B digitizer and Clampfit 10.5 (Molecular Devices).

## Immunohistochemistry

Mice were transcardially perfused with 20 ml saline, followed by 20 ml 4% (weight divided by volume; wt/vol) phosphate buffer saline (PBS)-buffered paraformaldehyde under deep anesthesia with 10% chloral hydrate (400 mg/Kg). Brains were dissected and postfixed overnight at 4˚C, then treated with 30% sucrose at 4˚C for 2 days. Coronal sections with thickness of 30 μm were cut with Leica VT1200 Semiautomatic Vibrating Blade Microtome. Sections were permeabilized with PBS-T (0.1% Triton 100X, Sigma, United Kingdom) and blocked in PBS-T including 5% (wt/vol) normal donkey serum (Sigma) for 1 h. For labelling LHb neurons, the sections were incubated in the primary antibody rabbit anti-NeuN (1:500; 26975-I-AP, ProteinTech, UK) diluted in PBS-T with 5% normal donkey serum at 4˚C with gentle shaking for 3 days, then washed 3 times in PBS-T. For visualization, anti-NeuN was detected by the secondary antibody goat anti-Rabbit Alexa 488 (1:500; 711-545- 152, Jackson ImmunoResearch Laboratories, Pennsylvania, USA), and all sections were counter-stained with DAPI (300 nM; final concentration, Sigma) for 2 days' staining at 4˚C with gentle shaking. Sections were then washed and mounted in PBS on Polysine microscope slides (VWR International, LLC, Pennsylvania, USA), air-dried overnight, and coverslipped with Aquatex aqueous mounting agent (VWR International, LLC). For labelling DRN GABAergic interneurons C56BL/6J mice received 100 nl of AAV2.1-mDlx-GFP (Addgene #83900) injection into the DRN (in mm: anterior/posterior, −4.7; lateral/medial, 1.1; dorsal/ventral, −3.2). To label LHB cells 100 nl of AAV2.1-hSyn-(H134R)-mCherry (Addgene #26976) was injected into the LHb (in mm: anterior/posterior, −1.6; lateral/medial,1.1; dorsal/ventral, −3.1). The mice were killed 2 months later, perfused with 4% PFA in PBS and postfixed overnight at 4˚C. The brains were sectioned at 40 μm thickness and floating stained with the following primary antibodies: Chicken anti GFP (Aves Labs, GFP-1020), Rabbit anti mCherry (abcam, ab167453), and Rat anti serotonin (abcam, ab6336). The secondary antibodies were: Goat anti-Chicken Alexa 488 (Thermo Fisher Scientific, A11039), Donkey Anti-Rabbit Cy3 (Jackson ImmunoResearch Laboratories, 711-165-152), Donkey Anti-Rat Alexa 647 (Jackson ImmunoResearch Laboratories, 712-606- 150). The sections were mounted and coverslip with ProLong Diamond Antifade Mountant (Thermo Fisher Scientific, P36961). All images were either (i) captured on Olympus FV1000 confocal microscope (Olympus Corporation, Japan) at 10× magnification or (ii) Leica DM6000 SP8 with 63× objective in single optical slice at the New York University Abu Dhabi Microscope Core facility and analyzed with Olympus FV1000 software.

## Statistical analysis

The estimated sample sizes were based on our past experience performing similar experiments and power analysis. Animals were randomly assigned to treatment groups. Analyses were

performed in a manner blinded to treatment assignments in all behavioural experiments. Data are presented as mean ± SEM. All analysis was performed using GraphPad Prism software v7 for Windows (La Jolla, California). By preestablished criteria, values were excluded from analyses if the viral injection sites were out of the LHb. When comparing 2 groups of normally distributed data, unpaired Student two-tailed $t$ test was used. Multiple-groups comparisons were achieved by means of analysis of variance with/without repeated factors (one-way or two-way ANOVA), followed by a post hoc Dunnett/Tukey multiple comparisons test. Two criteria were used to confirm burst firing: (1) Onset was defined as spikes beginning with a maximal interspike interval of 20 ms and ended with a maximal interspike interval of 100 ms, and minimum intraburst interval is 100 ms, and (2) minimum number of spikes in a burst is 2. All statistical tests were two-tailed, and statistical significance was set at $P < 0.05$.

## Supporting information

**S1 Fig. Relates to Fig 1. Social stress induces social avoidance in susceptible mice. (A)** SI data showed that in the presence of a CD1 social target (nonaggressor), susceptible mice displaying: **(i)** decreased time in the interaction zone ($F_{2,9} = 5.185$, $P < 0.05$), **(ii)** increased time in the corner zone ($F_{2,9} = 5.188$, $P < 0.05$), and **(iii)** no difference in total travel between control, susceptible, and resilient mice. **(B)** Though not significant, correlation analysis of in vivo spontaneous activity in the LHb and SI ratio showed a very slight negative correlation. Error bars: mean ± SEM. The raw data can be found in S7 Data. LHb, lateral habenula; SI, social interaction.
(TIF)

**S2 Fig. Relates to Fig 2. SI data in mice used to measure day or nighttime spontaneous firing in cells$^{LHb \rightarrow DRN}$. (A)** Susceptible mice used for day or **(B)** nighttime in vitro recording of labelled cells$^{LHb\text{-}DRN}$ display: decreased SI ratio (day—$F_{2,24} = 28.02$, $P < 0.0001$; $n = 7\text{--}11$ mice/group; night—$F_{2,12} = 8.959$, $P < 0.01$; n = 4–6 mice/group) (left). In the presence of CD1 (social target) susceptible mice spent increased time in the corner zone (day—$F_{2,24} = 3.335$, $P < 0.05$; night—$F_{2,12} = 6.444$, $P < 0.05$) (middle). There was no difference in total travel between control, susceptible, and resilient mice (right). **(C)** The in vitro electrophysiology experiments were performed over 16 days after the SI test. To better visualize the rhythmic changes in spontaneous firing in cells$^{LHb \rightarrow DRN}$, data were regraphed where day and night firing was binned into first half (1–8 Day/Night after SI) and the second half (9–16 Day/Night after SI) recording sessions. In susceptible mice, the day time firing pattern is phase inversed and was significantly higher on day 1 than control and resilient mice (Day 1 –control vs susceptible: $F_{2,160} = 3.617$, $P = 0.0306$; resilient vs susceptible: $F_{2,160} = 3.689$, $P = 0.0268$; $n = 53\text{--}64$ cells from 7 to 11 mice/group), and day 2 from control mice (Day 2 –$F_{2,160} = 3.653$, $P = 0.0286$), ***day vs day in control vs susceptible and resilient vs susceptible mice, **control vs susceptible. Error bars: mean ± SEM. The raw data can be found in S8 Data. SI, social interaction.
(TIF)

**S3 Fig. Relates to Fig 2. Diurnal difference in intrinsic membrane properties in mice exposed to CSDS. (A)** Sample traces of evoked firings of cells$^{LHb\text{-}DRN}$ in day (top left) or night (bottom left) in control (top), susceptible (middle), and resilient mice (bottom). Cells$^{LHb\text{-}DRN}$ from susceptible mice display increased daytime excitability in response to incremental steps in current injections (75, 100, 125, 150, 175, and 200 pA) compared with control and resilient mice ($F_{2,820} = 21.61$, $P < 0.0001$; $n = 14$ to 42 cells from 7 to 11 mice per group) (top right). There was no difference between the stress phenotypes at night (bottom right). **(B)** Cells$^{LHb\text{-}}$

DRN from susceptible mice display decreased daytime rheobase ($F_{2,82}$ = 14.37, $P$ < 0.0001; $n$ = 22 to 38 cells from 7 to 11 mice per group) (top). There was no difference in nighttime rheobase between the stress phenotypes (bottom). **(C)** Representative voltage traces in response to hyperpolarizing current injections in cells^LHb-DRN in day (top left) or night (bottom left) in control (top), susceptible (middle), and resilient mice (bottom). Daytime $I$–$V$ relationship showing cells^LHb-DRN from susceptible mice display increased membrane resistance ($F_{2,264}$ = 23.13, $P$ < 0.0001; $n$ = 9 to 15 cells from 7 to 11 mice per group) (top right). There was no difference in Nighttime $I$–$V$ relationship between the stress phenotypes (bottom right). Error bars: mean ± SEM. The raw data can be found in S9 Data. CSDS, chronic social defeat stress. (TIF)

**S4 Fig. Relates to Fig 3. Effects of HCN channel and $K_V^+$-channel antagonists on spontaneous activity. (A)** Pie charts illustrating the percentage of cells excited or inhibited by HCN antagonist ZD7288 (10 μM). **(B, C)** One-way ANOVA does not show significant difference in the % excitation or inhibition of spontaneous firing between control, resilient, and susceptible mice in the day or the night. (Day % Inhibition: $n$ = 5–10 cells; Day % Excitation: $n$ = 6–12 cells; Night % Inhibition: $n$ = 8–12 cells; Night % Excitation: $n$ = 9–15 cells from 2–3 mice/ group). **(D)** Pie charts illustrating the percentage of cells excited or inhibited by $K_V^+$-channel antagonist TEA (10 mM). **(E)** One-way ANOVA shows significant difference in the % inhibition in the day ($F_{2,27}$ = 3.42, $P$ = 0.047, $n$ = 8–12 cells from 2–3 animals/group). Post hoc analysis showed significant % inhibition in susceptible mice compared to control mice (Tuckey multiple comparison: $P$ = 0.001). There was no significant difference in % inhibition at night ($n$ = 9–11 cells from 2–3 animals/group). **(F)** One-way ANOVA shows significant difference in % excitation in the night ($F_{2,26}$ = 4.63, $P$ = 0.02, $n$ = 7–15 cells from 2–3 animals/group). Post hoc analysis showed significant % excitation in susceptible mice compared to resilient mice (Tuckey multiple comparison: $P$ = 0.02). There was no statistical difference in % excitation in the day ($n$ = 7–12 cells from 2–3 animals/group). The raw data can be found in S10 Data. HCN, hyperpolarization-activated cation; TEA, tetraethylammonium. (TIF)

**S5 Fig. Relates to Fig 4. Functional validation of AAV-DIO-ChR2-mCherry and optical stimulation after each exposure to social stress during the SDS paradigm. (A)** Experimental timeline of pathway-specific in vitro recording of labelled cells^LHb-DRN (top), and schematic showing surgeries site for virus injections to specifically label cells^LHb-DRN (bottom). **(B)** Schematic showing the optical fiber used for in vitro delivery of blue light (470 nm) and glass electrode used for simultaneous whole-cell recording of labelled cells^LHb-DRN responses (top left). Whole-cell current-clamp recordings from AAV-DIO-ChR2-mCherry-infected cells^LHb-DRN in LHb slices showing 20 Hz_20 ms of blue light stimulation induces tonic firings (middle left). Whole-cell voltage clamp recordings showing long duration (2 s) of blue light stimulation induces temporally precise inward photocurrent (bottom left), Frequency-response curve of membrane excitability to blue light stimulation showing light frequencies ranging from 0.1–20 Hz reliably induce the approximately equivalent firing rate in the ChR2-mCherry expressing cells^LHb-DRN (right). **(C)** Detailed schematic of the SDS procedure during which the ChR2 expressing cells^LHb-DRN were optically stimulated for 20 min after each exposure to social stress for 7 days, after which the mice underwent the SI test. **(D)** Cartoon showing in vivo optical stimulation protocol during the SDS paradigm. **(E)** In the presence of a CD1 social target (nonaggressor), SDS-ChR2 mice display increased time in the corner zone ($F_{2,18}$ = 3.55, $P$ < 0.05) (right). There was no difference in total travel (right). $N$ = 5–7 mice/group. Error bars: mean ± SEM. The raw data can be found in S11 Data. LHb, lateral habenula; SDS, social

defeat stress; SI, social interaction.
(TIF)

**S6 Fig. Relates to Fig 5. No difference in endogenous rhythms in mice exposed to CSDS.**
**(A)** In the presence of a CD1 social target (nonaggressor), susceptible mice displaying **(i)**
decreased SI ratio ($F_{2,27}$ = 18.08, $P < 0.0001$), **(ii)** decreased time in the interaction zone ($F_{2,27}$
= 19.68, $P < 0.0001$), **(iii)** increased time in the corner zone ($F_{2,27}$ = 5.188, $P < 0.05$), **(iv)** no
difference in total travel. **(B)** Representative double-plotted actograms of control, susceptible,
and resilient mice. Grey background indicating either dark phase of the LD cycle or DD; light
blue and pink lines representing extended regression line derived by onset of activity under
LD and DD cycle, respectively. **(C–F)** No difference in (C) phase change in activity onset in
DD, (D) in correlation between interaction ratio and degree of phase change, (E) in free run-
ning period length in DD, and (F) in total activity counts in 24 h between LD and DD between
control, susceptible, and resilient mice. Though not significant, susceptible mice exhibit
slightly larger phase change (C) and shorter free running period (E). **(G, H)** Intradaily variabil-
ity was lower in stress-exposed mice compared to stress-naïve groups ($P < 0.05$). (I, J) There
was no difference in interdaily stability between the 3 phenotypes. $N$ = 8–12 mice/group. Error
bars: mean ± SEM. The raw data can be found in S12 Data. DD, complete darkness; CSDS,
chronic social defeat stress; LD, light–dark; SI, social interaction.
(TIF)

**S7 Fig. Relates to Fig 6. LHb projections innervate both serotonergic and GABAergic neu-**
**rons in the DRN. (A)** Immunohistochemical staining in DRN showing mCherry (red) (A)
that represents axons projecting from the LHb infected with AAV2.1-hSyn-hChR2(H134R)-
mCherry. The enriched dotted staining on the soma for both GFP-positive GABAergic neu-
rons (B in green as indicated by arrows), and 5HT-positive serotonergic neurons (C in blue).
GFP expression was driven by GABAergic-specific mDlx enhancer in DRN infected with
AAV2.1-mDlx-GFP. The 3 colours were merged in D. The amount of red dots on the blue
serotonergic cells are compatible to those on the green GABAergic cells, suggesting that LHb
projections into DRN likely innervate to both DRN GABAergic and DRN 5-HT cells. (B) The
in vitro electrophysiology experiments were performed over 16 days after the SI test. To better
visualize the rhythmic changes in spontaneous firing in cells$^{LHb \rightarrow DRN}$, data were regraphed
where day and night firing was binned into first half (1–8 Day/Night after SI) and the second
half (9–16 Day/Night after SI) recording sessions. DRN cells display daily rhythms in sponta-
neous firings in resilient mice with significantly higher daytime activity than night (Day 1—
$F_{2,79}$ = 4.125, $P$ = 0.0234; Day 2—$F_{2,79}$ = 4.111, $P$ = 0.0240; $n$ = 53–64 cells from 4 to 10 mice/
group). Also, in daytime, spontaneous firing of DRN cells was significantly higher in resilient
mice than susceptible (Day 1—$F_{3,79}$ = 4.085, $P$ = 0.0137; Day 2—$F_{3,79}$ = 4.072, $P$ = 0.0141;
$n$ = 53–64 cells from 4 to 10 mice/group). $^*$day vs night in resilient mice, #day vs day in resil-
ient vs susceptible mice. Error bars: mean ± SEM. The raw data can be found in S13 Data.
DRN, dorsal raphe nucleus; GFP, green fluorescent protein; LHb, lateral habenula; SI, social
interaction.
(TIF)

**S1 Data. Metadata for Fig 1D and 1E(iii).**
(XLSX)

**S2 Data. Metadata for Fig 2D, 2E, 2F and 2G.**
(XLSX)

**S3 Data. Metadata for Fig 3A, 3B, 3C and 3D.**
(XLSX)

**S4 Data. Metadata for Fig 4B and 4C.**
(XLSX)

**S5 Data. Metadata for Fig 5C and 5D.**
(XLSX)

**S6 Data. Metadata for Fig 6B, 6C, 6D and 6E.**
(XLSX)

**S7 Data. Metadata for S1A(i), S1A(ii), S1A(iii) and S1B.**
(XLSX)

**S8 Data. Metadata for S2A, S2B and S2C.**
(XLSX)

**S9 Data. Metadata for S3A, S3B and S3C.**
(XLSX)

**S10 Data. Metadata for S4A, S4B, S4C, S4D, S4E and S4F.**
(XLSX)

**S11 Data. Metadata for S5B and S5E.**
(XLSX)

**S12 Data. Metadata for S6A(i-iv), S6C, S6D, S6E and S6F.**
(XLSX)

**S13 Data. Metadata for S7B.**
(XLSX)

## Author Contributions

**Conceptualization:** He Liu, Ashutosh Rastogi, Basma Radwan, Dipesh Chaudhury.

**Data curation:** He Liu.

**Funding acquisition:** Dipesh Chaudhury.

**Investigation:** He Liu, Ashutosh Rastogi, Priyam Narain, Qing Xu, Merima Sabanovic, Ayesha Darwish Alhammadi, Lihua Guo, Hala Aqel, Vongai Mlambo, Rachid Rezgui, Basma Radwan.

**Project administration:** Dipesh Chaudhury.

**Resources:** Jun-Li Cao, Hongxing Zhang.

**Supervision:** Dipesh Chaudhury.

**Writing – original draft:** Dipesh Chaudhury.

**Writing – review & editing:** He Liu, Ashutosh Rastogi, Merima Sabanovic.

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
