## [Editor Report · Decision Letter 0]

7 Mar 2020

Dear Dr Chaudhury, 

Thank you for submitting your manuscript entitled "Blunted Diurnal Firing in Lateral Habenula Projections to Dorsal Raphe Nucleus and Delayed Photoentrainment in Stress-Susceptible Mice." for consideration as a Research Article by PLOS Biology.

Your manuscript has now been evaluated by the PLOS Biology editorial staff as well as by an academic editor with relevant expertise and I am writing to let you know that we would like to send your submission out for external peer review.

Please re-submit your manuscript within two working days, i.e. by Mar 09 2020 11:59PM.

Kind regards,

Di Jiang

PLOS Biology

---

## [Decision Letter · Decision Letter 1]

8 Apr 2020

Dear Dr Chaudhury,

Thank you very much for submitting your manuscript "Blunted Diurnal Firing in Lateral Habenula Projections to Dorsal Raphe Nucleus and Delayed Photoentrainment in Stress-Susceptible Mice." for consideration as a Research Article at PLOS Biology. Your manuscript has been evaluated by the PLOS Biology editors, an Academic Editor with relevant expertise, and by four independent reviewers.

In light of the reviews (below), we will welcome re-submission of a much-revised version that takes into account the reviewers' comments. Our Academic Editor has examined carefully all the comments and provided you with a list of essential experiments and clarifications in revision.

"Reviewer one:

Points one and two are both important and need to be addressed in a point-by-point response. In particular, I would like to hear the author’s explanation as a response for both comments to a possible diurnal rhythm in SI scores and the correlation with the LHb firing rate.

Point three is essential. This would be a required experiment: The authors should use the inhibitor suggested by the reviewer as this could allow the authors’ to have more mechanistic insights about the process.

Point four is minor and can be addressed by the authors through simple clarification of the process.

Point 5 is another crucial experiment: That is to say to show what the LHb stimulation do to the DRN neurons. A wonderful addition would be to also carry out point 6, as this would add impressively to the manuscript. However, point 6 can be the subject of future studies.

The entrainment data is exciting and is discussed quite well in context. I believe including some data on the role of 5HT on SCN light responsiveness will also help to keep this data in this manuscript. As DRN neurons also innervate the SCN.

Reviewer two:

Point one should be addressed in a point-by-point response.

Point two will be answered by point 3 experiments from reviewer 1

Point three is important, but beyond the scope of this study. Should be addressed in a point-by-point response.

Point 4 will be addressed by doing the experiments for point 5 from reviewer 1.

Ponit 5 should be addressed in a point-by-point response.

Reviewer three:

Point 1 should be analyzed as it is also mentioned by reviewer 1. This is concerning the inversion of the rhythm.

Comments on point 2 and non-photic entrainment should be added.

Point 3 could be helpful, but up to the authors to provide or not.

Reviewer four:

The authors should consider the sole suggestion provided by this reviewer."

We cannot make any decision about publication until we have seen the revised manuscript and your response to the reviewers' comments. Your revised manuscript is also likely to be sent for further evaluation by the reviewers. We expect to receive your revised manuscript within 2 months. 

**IMPORTANT - SUBMITTING YOUR REVISION**

*Re-submission Checklist*

*Published Peer Review*

*PLOS Data Policy*

*Blot and Gel Data Policy*

Sincerely,

Di 

PLOS Biology

REVIEWS:

Reviewer #1 (Mario A. Penzo, signed review): In their manuscript entitled "Blunted Diurnal Firing and Delayed Photoentrainment in Stress-Susceptible Mice" Liu and colleagues use the chronic social defeat stress paradigm (CSDS) to show that susceptibility to stress is associated with changes in the spontaneous firing rates of LHb neurons that project to the DRN. In addition, the authors show that in naïve and resilient mice, LHb-DRN neuron display diurnal variations in firing rates. Importantly, this daily rhythm is blunted in susceptible mice. The authors also explored the potential contribution of specific conductances to diurnal variations in firing, but from the authors' perspective the results from these experiments seem counter-intuitive and somewhat difficult to interpret. However, I believe that some of their results in this category should be explored further, as they could offer insight on the potential mechanisms underlying susceptibility (more on this below). The authors also show that stress (both susceptibility and resiliency) is associated with changes in entrainment to light advances and that these changes are different in susceptible and resilient subjects. Finally, the authors attempt to draw parallels between susceptibility-associated changes in spontaneous firing rates in the LHb and those observed in the DRN. These later aspects of the study (entrainment and DRN electrophysiology), however, are substantially underdeveloped. As a result, the Discussion section is highly throughout. This leaves the reader with the impression that drawing concrete conclusions from the result presented here is rather difficult. As such, while I find that the study, and in particular its core finding, would be of interest to a wide scientific audience, some critical data is missing here. In particular, the author's need to strengthen the mechanistic aspects of their study, by causally linking cellular and circuit mechanisms to behavior. It is the reviewer's impression that this is a reasonable expectation that should be met for publication in PLOS Biology. Below is a point-by-point review with critiques and recommendations:

1) The authors mention that SI scores for control, resilient and susceptible mice were comparable at day and night times. However, it seems that some diurnal variability exists in the naïve group, with animals at night showing higher SI scores. Are the behavior tests (CSDS) and SI also occurring at different ZT times? I am trying to figure out if only the recordings happen at this time or if behavioral manipulations are similarly restricted in a diurnal fashion. If they are, then the behavioral effects observed in the naïve group seems contradictory because LHb neurons have higher firing rates at night. Have the authors run comparisons between these two groups to see if there are statistically significant differences in naïve groups? Given that the N is low for these behavioral tests (since these are subjects used for electrophysiological studies) more data will be needed to achieve more statistical power.

2) Data in Figure 2 demonstrate that the firing rates of susceptible neurons do not fluctuate as much, as the abstract suggests. But, unlike naïve and resilient mice (in which firing is higher during daytime), firing rate in susceptible seems larger during daytime compared to nighttime. So, diurnal variations in LHb neurons of susceptible mice are not only blunted but inverted. This is an important result, that is only briefly mentioned in the figure legend of Supplementary Figure 2, but mostly ignored throughout the study. What is the reason for this inversion?

3) I was particularly surprised that the authors largely neglected their I-h results. While I-h conductance doesn't show diurnal rhythms in LHb-DRN neurons, stress seems to increase I-h in both susceptible and resilient mice, but the effect is apparently larger in susceptible subjects. Given that I-h is known to contribute to increased excitability, for example in the adjacent paraventricular nucleus of the thalamus (Kolaj et al., 2012; Journal of Neurophysiology), this conductance is a major suspect for the marked increase in daytime firing rates displayed by LHb neurons of susceptible mice. The authors could assess the contribution of this conductance from neurons across all three conditions at both day- and nighttime. This can be achieved by comparing the effect of the I-h inhibitor ZD7288 at both times of the cycle. In addition, the authors could attempt genetic manipulations targeting I-h to investigate the relationship between increased I-h and stress susceptibility. I am not entirely sure why the authors haven't explored this avenue further, given the senior author's previous research on this topic.

4) As mentioned above, a major limitation of this work is the lack of manipulations to link individual findings. This makes the paper feel as composed of attractive vignettes that are not linked into a cohesive story. For example, data presented in Figure 5 shows that stress affects circadian rhythms but that susceptible and resilient mice show somewhat opposing phenotypes. So, while susceptible mice show initial delay in entrainment but ultimately entrain in a similar timeframe as controls, resilient mice seem to entrain significantly faster than controls. This result is interesting when treated as an independent observation. However, there is no evidence linking LHb activity to these different phenotypes. As such, it is unclear that the stress induced firing phenotypes seen in LHb and explored in Figures 1 and 2 are related to the circadian phenotype. 

5) A similar issue arises when evaluating the DRN data. While this data is very interesting on its own, the parallels drawn between decreased spontaneous firing in the DRN and increases firing in the LHb in susceptible mice is only speculative. Indeed, the authors suggest that feedforward inhibition of DRN neurons resulting from increased LHb-DRN activity is the most parsimonious interpretation of the susceptible state. But evidence that this is the case is missing. Given the authors' technical expertise I am confident that this is something they can address experimentally. Why leave it to speculation?

6) Related to the previous point, are the DRN effects exclusive to 5-HT neurons? The authors should consider selectively assessing these effects while contrasting GABA and 5-HT DRN neurons. 

Overall, I believe that addressing these issues will allow the authors to achieve a more rounded study that doesn't seem so speculative. One possibility (if other reviewers agree) would be to remove the entrainment data and focus on linking the LHb and DRN phenotypes. In addition, as stated above, the authors must address the possibility that some of the identified conductances, in particular I-h, play a fundamental role on stress susceptibility. 

Minor:

1) Consider combining Figure 1 and Supplementary Figure 1.

2) Also, consider citing the Kolaj et al., 2012 (Leo Renaud's lab) study about diurnal variations in spontaneous firing in the neighboring paraventricular thalamus. That study found that, unlike in the LHb (present stud), diurnal changes in I-h occur in naïve mice. 

Reviewer #2: In this study, Liu et al. report that the excitability of DRN-projecting LHb cells in stress-susceptible mice is significantly increased during the day, while the excitability of DRN cells is significantly reduced. They also find that specific activation of the DRN-projecting LHb cells during the day induces stress-susceptibility. Finally, they report that stress induces blunted daily rhythms in the firing of DRN-projecting LHb cells. Although the experiments appear to be well-controlled and interpretable, there are large number of technical and conceptual issues that suggest that many of the present experiments are not up to the standards of this journal. 

Major points:

1. Previous studies found that exposure to stress significantly increased burst firing in LHb cells (Yang et al., 2018; Huang et al., 2019). In contrast, the authors found that exposure to CSDS significantly decreased burst firing in LHb cells (Fig. 2H). The authors should explain this apparent discrepancy. 

2. The authors reported that susceptible mice showed altered Ih-current and Kv+ currents in DRN-projecting LHb cells, accompanied by elevated diurnal, spontaneous firing. However, the causal relationships between the changes of Ih-current/Kv+ currents and the spontaneous firing of LHb cells are still lacking. 

3. The authors found that specific activation of DRN-projecting LHb cells induced stress-susceptibility. It is possible that DRN-projecting LHb cells may also project to mood-related brain regions other than the DRN. To further confirm whether the LHb-DRN pathway is implicated in stress-susceptibility, the authors should regulate neuronal activity of DRN that can receive direct input from the LHb and exam its effects on depressive-like behaviors. 

4. The authors found that the spontaneous firing of DRN-projecting LHb cells was significantly increased, while the spontaneous firing of DRN cells was significantly decreased. It is well documented that most of LHb cells are excitatory. Why the enhanced excitatory inputs to the DRN can reduce the spontaneous firing of DRN cells? What is the identity of the recorded DRN? Are they directly innervated by the LHb?

5. The authors performed in vivo recordings in anesthetized mice. Anesthesia may affect firing of neurons (Schonewille et al., 2006). The authors should conduct the in vivo recording experiments in behaving animals.

Minor point:

1. The authors should explain why they choice tonic stimulation instead of phasic stimulation in the optogenetic stimulation experiment.

2. There are a few typos throughout the manuscript. The authors should carefully proof the manuscript.

References:

1. Yang Y, et al. Ketamine blocks bursting in the lateral habenula to rapidly relieve depression. Nature. 2018 Feb 14;554(7692):317-322.

2. Huang L, et al. A Visual Circuit Related to Habenula Underlies the Antidepressive Effects of Light Therapy. Neuron. 2019 Apr 3;102(1):128-142.e8.

3. Schonewille M, et al. Purkinje cells in awake behaving animals operate at the upstate membrane potential. Nat Neurosci. 2006 Apr;9(4):459-61;

Reviewer #3: This paper describes a detailed and very elegant series of experiments exploring the role of dorsal raphe nuclei (DRN) projecting neurons of the lateral habenula (LHb) in a chronic social defeat stress (CSDS) mouse model of depression. The key findings are that stress-susceptible mice show a blunting of diurnal firing patterns in the LHb>DRN pathway, and that even sub-threshold stressed animals can be made to show depressed behaviour via stimulation of this circuit. Furthermore, the authors show that disruption of this circuit in this mouse model of depression is associated with altered circadian photoentrainment. These data provide important insight into the links between the circadian system and specific neuronal circuits in mood disorders.

Overall, the paper is very clearly presented and progresses logically. The authors first characterise LHb firing, then examine the specific Lhb>DRN projections, before characterising the role of specific channels. Via the use of detailed electrophysiological recordings, the authors identify a role for HCN channels and sustained Kv+ channels in response to CSDS. They then show use optogenetic studies on DRN projecting LHb neurons in subthreshold SDS (SSDS) mice, showing that stimulation of these cells is sufficient to produce stress-susceptibility. They then show that whilst circadian rhythms appear intact, that resilient mice show faster re-entrainment to a jet lag style shift in the LD cycle. Finally, they show rhythms in DRN spontaneous firing are also blunted in stress susceptible mice.

The links between the habenula, mood and photoentrainment are currently highly topical given high profile studies on the role of light in regulating mood and the role of the habenula (Fernandez et al., 2018 Cell; Huang et al., 2019 Neuron). This study provides critical insight into the links between mood disorders and circadian disruption, and the characterisation of the LHb/DRN/SCN circuit will be of interest to many research fields.

Specific comments are provided below:

1. In several places in the paper, changes in spontaneous neuronal activity are compared between control, stress susceptible and stress resilient animals during the day or night. For example, in Figure 2D-E, Lhb>DRN neurons show an increase in daytime firing in stress susceptible mice, but no differences occur during the night. I was surprised that day vs night are not explicitly compared, as this could be informative. For example, in the data in Figure 2D-E, it looks like the normal (control) mice show low firing during the day and higher on a night. A similar pattern is observed in the stress resilient mice. By comparison, the stress-susceptible mice show high firing during the day and lower firing on a night. This suggests that the rhythms in these cells are not just blunted, but actually inverted. I appreciate that there may be technical reasons why direct comparison between electrophysiological data may not be directly compared between preparations, but this may be an interesting observation.

2. Whilst stress-resilient mice show faster re-entrainment, are stress-susceptible mice actually slower at re-entraining? These seem comparable with controls. Could this faster re-entrainment be as a result of differences in non-photic entrainment in stress resilient mice?

3. It may be helpful to the reader to provide a figure showing the proposed interactions in the LHb, DRN and SCN circuit, and how these are affected in depression.

Reviewer #4: The study by Liu et al investigates the role of the lateral habenula neurons that project to dorsal raphe in the impact of stress on circadian rhythms. The authors show that LHB-DRN cells display blunted diurnal firing in mice susceptible to social defeat stress compared to control or resilient conditions. They further show that enhancing activity in LHB-DRN cells using optogenetics during social defeat stress induces social avoidance. Finally, the authors demonstrate that susceptible mice display slower photoentrainment to a shift in LD cycle while resilient mice display a faster photoentrainment. Overall this is an important and rigorous study that provides new information into the circuitry involved in stress effects on circadian rhythms. I only have one minor suggestion below that would help to improve the clarity of the study.

This reviewer understands that SDS protocols are modified across labs based on what is optimal in each laboratory. However, the 7 day SDS would not typically be considered a subthreshold SDS. Perhaps the authors could simply modify their nomenclature to 15D-SDS and 7D-SDS and explain why the 7 days was advantageous for the optogenetic susceptible inducing studies.

---

## [Decision Letter · Decision Letter 2]

15 Jan 2021

Dear Dr Chaudhury,

Thank you for submitting your revised Research Article entitled "Blunted Diurnal Firing in Lateral Habenula Projections to Dorsal Raphe Nucleus and Delayed Photoentrainment in Stress-Susceptible Mice." for publication in PLOS Biology. I have now obtained advice from the original reviewers and have discussed their comments with the Academic Editor. You will note that reviewer 1, Mario A Penzo, reveals his identity.

As you will see, reviewers 1 and 2 are not yet persuaded that your data fully support your conclusion. However, in agreement with the Academic Editor, we think that a minor textual revision that more clearly highlights the limitations of your findings is publishable in PLOS Biology. Therefore, we will probably accept this manuscript for publication, assuming that you will modify the manuscript to address the remaining points raised by the reviewers by including the following or an equivalent paragraph at the beginning of your discussion: "Although we labeled the cells that project to the DRN from the LHb, the possibility still exists that these same LHb neurons project to sites distinct from the DRN that may be responsible for the mood effects that we observe". 

Please also make sure to address the data and other policy-related requests noted at the end of this email.

We expect to receive your revised manuscript within two weeks. Your revisions should address the specific points made by each reviewer. 

-  a cover letter that should detail your responses to any editorial requests, if applicable

*Published Peer Review History*

*Early Version*

Sincerely,

Gabriel Gasque, Ph.D.,

Senior Editor,

ggasque@plos.org,

PLOS Biology

DATA POLICY:

Note that we do not require all raw data. Rather, we ask for all individual quantitative observations that underlie the data summarized in the figures and results of your paper. For an example see here: http://www.plosbiology.org/article/info%3Adoi%2F10.1371%2Fjournal.pbio.1001908#s5

These data can be made available in one of the following forms:

Regardless of the method selected, please ensure that you provide the individual numerical values that underlie the summary data displayed in the following figure panels: Figures 1DE, 2DEFGH, 3ABCD, 4BC, 5CD, 6BCDE, S1AB, S2ABC, S3ABC, S4ABCDEF, S5BE, S6ACDEFGHI, and S7B.

Please also ensure that each figure legend in your manuscript includes information on where the underlying data can be found and that your supplemental data file/s has/have a legend.

Reviewer remarks:

Reviewer #1, Mario A Penzo: In their revised manuscript and accompanying rebuttal letter the authors have responded to my initial critiques. However, while the authors' response to some of my inquiries are appropriate, important requested experimental work is still missing here. The authors have offered somewhat reasonable justifications for this, which include technical limitations. But unfortunately, without additional experimental insight, I am not confident that the paper meets the standard for publication in PLOS Biology.

Reviewer #2: Although the authors have addressed some of my comments, and the manuscript has improved, I still have a few concerns regarding the specificity of this work. 

1.As I pointed out in my previous review, activation of DRN-projecting LHb neurons may influence the activity of brain regions other than the DRN, which can also receive innervation from the LHb. The authors did not answer my question directly. To convince the readers that the depression-inducing effects observed by activation of DRN-projecting LHb cells is mediated by the LHb-DRN pathway, the authors should show the underlying projection targets of DRN-projecting LHb cells, and provide evidence that only specific activation of the LHb-DRN pathway could induce stress-susceptibility. 

2. Although I understand that the epidemic has had a serious impact on the experiment, I still think the authors should provide data (at least some morphological data) to help the reader understand why activation of LHb cells inhibits DRN 5-HT cells.

Reviewer #3: 

The authors have addressed all of my comments in the revised manuscript.

---

## [Editor Report · Decision Letter 3]

25 Jan 2021

Dear Dr Chaudhury,

Thank you for submitting your revised Research Article entitled "Blunted Diurnal Firing in Lateral Habenula Projections to Dorsal Raphe Nucleus and Delayed Photoentrainment in Stress-Susceptible Mice." for publication in PLOS Biology. I writing because some of the metadata remains incomplete. 

As mentioned in my previous decision letter, please use the following format verbatim for your Data File: S1 Data

Please also ensure that each figure legend in your manuscript includes information on where the underlying data can be found (S1 Data) and that your supplemental data file has a legend.

Also, update your S1 Data file to include the following information: for Figures 3 and S3, the numerical data provided should include all replicates AND the way in which the plotted mean and errors were derived (it should not present only the mean/average values). Please also include data for Figure S5B.

We expect to receive your revised manuscript within two weeks. 

-  a cover letter that should detail your responses to any editorial requests, if applicable

*Published Peer Review History*

*Early Version*

Sincerely,

Gabriel Gasque, Ph.D.,

Senior Editor,

ggasque@plos.org,

PLOS Biology

---

## [Editor Report · Decision Letter 4]

2 Feb 2021

Dear Dr Chaudhury,

Thank you for submitting your revised Research Article entitled "Blunted Diurnal Firing in Lateral Habenula Projections to Dorsal Raphe Nucleus and Delayed Photoentrainment in Stress-Susceptible Mice." for publication in PLOS Biology. I am writing because some of the metadata remains incomplete.

As mentioned in my previous decision letter, please ensure your supplemental data files have legends.

We expect to receive your revised manuscript within two weeks.

To submit your revision, please go to https://www.editorialmanager.com/pbiology/ and log in as an Author. Click the link labelled 'Submissions Needing Revision' to find your submission record. If applicable, your revised submission must include the following:

- a cover letter that should detail your responses to any editorial requests, if applicable

*Published Peer Review History*

*Early Version*

Sincerely,

Gabriel Gasque, Ph.D.,

Senior Editor,

ggasque@plos.org,

PLOS Biology

---

## [Editor Report · Decision Letter 5]

4 Feb 2021

Dear Dr Chaudhury,

On behalf of my colleagues and the Academic Editor, Samer Hattar, I am pleased to say that we can in principle offer to publish your Research Article "Blunted Diurnal Firing in Lateral Habenula Projections to Dorsal Raphe Nucleus and Delayed Photoentrainment in Stress-Susceptible Mice." in PLOS Biology, provided you address any remaining formatting and reporting issues. These will be detailed in an email that will follow this letter and that you will usually receive within 2-3 business days, during which time no action is required from you. Please note that we will not be able to formally accept your manuscript and schedule it for publication until you have made the required changes.

PRESS

Thank you again for supporting Open Access publishing. We look forward to publishing your paper in PLOS Biology. 

Sincerely, 

Gabriel Gasque, Ph.D. 

Senior Editor 

PLOS Biology